# Analysis of one-hidden-layer Neural Networks via the Resolvent Method

**Vanessa Piccolo**
ETH Zurich (current affiliation: ENS Lyon)
vanessa.piccolo@ens-lyon.fr

**Dominik Schröder**
Institute for Theoretical Studies
ETH Zurich
dschroeder@ethz.ch

## Abstract

In this work, we investigate the asymptotic spectral density of the random feature matrix $M = YY^*$ with $Y = f(WX)$ generated by a single-hidden-layer neural network, where $W$ and $X$ are random rectangular matrices with i.i.d. centred entries and $f$ is a non-linear smooth function which is applied entry-wise. We prove that the Stieltjes transform of the limiting spectral distribution approximately satisfies a quartic self-consistent equation, which is exactly the equation obtained by Pennington and Worah [22] and Benigni and Péché [6] with the moment method. We extend the previous results to the case of additive bias $Y = f(WX + B)$ with $B$ being an independent rank-one Gaussian random matrix, closer modelling the neural network infrastructures encountered in practice. Our key finding is that in the case of additive bias it is impossible to choose an activation function preserving the layer-to-layer singular value distribution, in sharp contrast to the bias-free case where a simple integral constraint is sufficient to achieve isospectrality. To obtain the asymptotics for the empirical spectral density we follow the *resolvent method* from random matrix theory via the cumulant expansion. We find that this approach is more robust and less combinatorial than the moment method and expect that it will apply also for models where the combinatorics of the former become intractable. The resolvent method has been widely employed, but compared to previous works, it is applied here to non-linear random matrices.

## 1 Introduction

Machine learning has seen many successful achievements in recent years. Applications in face identification, object and speech recognition, translation, email spam filtering, navigation, medical diagnosis, etc. have proved the enormous potential of machine learning for day-to-day live [16, 11]. Deep neural networks have turned out to be a particularly powerful machine learning method, and understanding the theoretical underpinning of their success has received tremendous attention in mathematics, physics and computer science.

A fully-connected, feed-forward neural network with $L$ hidden layers of dimensions $n_1, \ldots, n_L$ can be modelled as follows:

$$f_\theta(\boldsymbol{x}) = \beta^* f(W^{(L)} f(W^{(L-1)} f(\ldots f(W^{(1)} \boldsymbol{x}) \ldots))) \in \mathbb{R}^d,$$

where $\boldsymbol{x} \in \mathbb{R}^{n_0}$ denotes the input data vector and $f \colon \mathbb{R} \to \mathbb{R}$ is a non-linear activation function which is applied entry-wise. We denote the parameters of the network by $\theta \coloneqq (W^{(1)}, \ldots, W^{(L)}, \beta)$, where $W^{(l)} \in \mathbb{R}^{n_l \times n_{l-1}}$ for $1 \leq l \leq L$ and $\beta \in \mathbb{R}^{n_L \times d}$ are the matrices of the weights. In the classical setting of supervised learning, we are given a training set of (say) $m$ samples of input feature vectors $\boldsymbol{x}_i \in \mathbb{R}^{n_0}$ with associated target vectors $\boldsymbol{z}_i \in \mathbb{R}^d$. For example, $\boldsymbol{x}_i$ may encode the pixels of a photograph of an animal and the target $\boldsymbol{z}_i$ may label the species of the

animal in the image. Roughly speaking, the goal of supervised learning is to learn the mapping between the feature and the target vectors based on a given training set in order to predict the output of new unlabelled data. Let $X = (\boldsymbol{x}_1 \ldots \boldsymbol{x}_m) \in \mathbb{R}^{n_0 \times m}$ be the matrix of the data and let $Z = (\boldsymbol{z}_1 \ldots \boldsymbol{z}_m) \in \mathbb{R}^{d \times m}$ be the target matrix. Then, the aim of the network is to find optimal parameters $\theta$ such that $f_\theta(X) = (f_\theta(\boldsymbol{x}_1), \ldots, f_\theta(\boldsymbol{x}_m)) \in \mathbb{R}^{d \times m}$ approximates the target $Z$ optimally. During the training phase, weights are adjusted in order to minimize the empirical risk $\mathcal{R}(\theta) = \mathbf{E}\,\mathcal{L}(f_\theta(X), Z)$, where $\mathcal{L}(\cdot, \cdot)$ is a given loss function, usually involving some penalty for large weights $\theta$ in order to avoid *over-fitting*. Stochastic gradient descent (SGD) and its variants with *back-propagation* are the most commonly used algorithms for training multilayer networks by iteratively updating the parameters into the direction of the negative of the gradient of the empirical risk. For a much more complete survey, we refer the reader to [11].

In the present paper, we will focus on a single-hidden-layer neural network of the form $f_\theta(X) = \beta^* Y$ with $Y = f(WX)$. This model was first studied by Louart, Liao, and Couillet [19] for the case where the data matrix $X$ is deterministic and $W$ is a matrix of random weights (in particular, the weights are given by functions of standard Gaussian random variables), and by Pennington and Worah [22] for the case where $X$ and $W$ are independent random matrices with both centred Gaussian entries. In both papers, the matrix $\beta \in \mathbb{R}^{n_1 \times d}$ is the only parameter to be learned and is chosen as the unique minimizer of the ridge-regularized least squares loss function

$$\mathcal{L}(f_\theta(X), Z) = \frac{1}{2dm}\|Z - \beta^* Y\|_F^2 + \gamma\|\beta\|_F^2,$$

where $\gamma > 0$ is the *learning rate*. The unique minimizing weight matrix $\hat\beta$ is then equal to $\hat\beta = YG(-\gamma)Z^*$, where

$$G(-\gamma) = \left(\frac{1}{m}Y^*Y + \gamma\right)^{-1}$$

is the resolvent of $\frac{1}{m}Y^*Y$. As proved in [19, 22], the expected training loss $E_{\text{train}}$ is related to $-\gamma\frac{\partial}{\partial\gamma}G(-\gamma)$, and thus also to the Stieltjes transform of the limiting spectral measure of $\frac{1}{m}Y^*Y$. Here the Stieltjes transform $m_\mu$ of a probability measure $\mu$ on $\mathbb{R}$ is defined as $m_\mu(z) := \int_{\mathbb{R}}(x-z)^{-1}\,\mathrm{d}\mu(x)$ for $z \in \mathbb{C}$ such that $\Im z \geq 0$, and for $\mu_{n_1}$ being the empirical probability measure of the $n_1$ eigenvalues of $\frac{1}{m}Y^*Y$ is related to the resolvent via $m_{\mu_{n_1}}(z) = \frac{1}{n_1}\operatorname{Tr} G(z)$. The performance of one-hidden-layer neural networks depends on the asymptotic spectral properties of the matrix $\frac{1}{m}Y^*Y$. Pennington and Worah [22] investigated the limiting spectral measure of the random matrix $M = \frac{1}{m}YY^*$ and derived the quartic self-consistent equation

$$
\begin{aligned}
1 + zg_\infty &= \theta_1(f)g_\infty\left(1 - \frac{\phi}{\psi}(1 + zg_\infty)\right) - \frac{\theta_2(f)}{\psi}g_\infty(1 + zg_\infty)\left(1 - \frac{\phi}{\psi}(1 + zg_\infty)\right) \\
&\quad + \frac{\theta_2(f)(\theta_1(f) - \theta_2(f))}{\psi}g_\infty^2\left(1 - \frac{\phi}{\psi}(1 + zg_\infty)\right)^2,
\end{aligned}
\tag{1}
$$

where $g_\infty(z) := \lim_{n_1 \to \infty} g(z)$ and $g(z) := \frac{1}{n_1}\operatorname{Tr} G(z)$ is the Stieltjes transform, which is approximately satisfied, $g_\infty(z) \approx g(z)$, by $g(z)$ in case of Gaussian $W, X$. It is notable that the asymptotic spectrum of $Y^*Y$ for large dimensions such that $n_0/m \to \phi \in (0, \infty)$ and $n_0/n_1 \to \psi \in (0, \infty)$ depends on the non-linear function $f$ only through two integral parameters $\theta_1(f)$ and $\theta_2(f)$, where $\theta_1(f)$ is the Gaussian mean of $f^2$ and $\theta_2(f)$ is the square of the Gaussian mean of $f'$ (c.f. (5)). Benigni and Péché [6] then extended this model to random matrices $W$ and $X$ with general i.i.d. centred entries, and obtained the same self-consistent equation (1). We mention that (1) may be reduced, for some special cases, to the quadratic equation that is satisfied by the Stieltjes transform $m_{\mu_{MP}}$ of the Marchenko-Pastur distribution $\mu_{MP}$ [20]. This means that for some activation functions, the non-linear random matrix model has the same limiting spectral distribution as that of sample covariance matrices $XX^*$ (in other cases, the equation can simplify to the cubic equation approximately satisfied by product Wishart matrices [7, 10]). This can be generalised to multilayer networks: [22] found experimentally that the singular value distribution is preserved through multiple layers by activation functions with $\theta_2(f) = 0$ and is given by the Marchenko-Pastur distribution in each layer. This conjecture was proved in [6] for the general case of bounded activation functions. Moreover, [19] performed a spectral analysis on the Gram matrix model with general training data and proved that, in the large dimensional regime, the resolvent of $Y^*Y$ has a similar behaviour as that

observed in sample covariance matrix models. This was extended in [17] by considering Gaussian mixture of data. We also refer to the recent paper [18]. In the context of multilayer feedforward neural networks, Fan and Wang [9] analysed the eigenvalue distribution of the Gram matrix model, where the weights are at random and the input vectors are assumed to be approximately pairwise orthogonal. In particular, they showed that the limiting spectral distribution converges to a deterministic limit and, at each intermediate layer, this limit corresponds to the Marchenko-Pastur map of a linear transformation of that of the previous layer.

In recent years, there has been some progress in the asymptotic analysis of the eigenvalue distribution of another Gram matrix, the so-called Neural Tangent Kernel (NTK). Consider a multilayer neural network and denote by $J = \nabla_\theta f_\theta(X)$ the Jacobian matrix of the network outputs with respect to the weights $\theta$. Then, the NTK is the Gram matrix of $J$, defined by $K^{\mathrm{NTK}} = J^*J$. It was shown in [14] that the NTK at random initialization converges, in the infinite-width limit, to a deterministic kernel and it remains constant during the whole training time of the network. Subsequently, [23] analysed the spectrum of the sample covariance matrix $JJ^*$ in a single-hidden-layer neural network, and provided an exact asymptotic characterization of the spectral distribution of $JJ^*$ with random Gaussian weights and data. Recently, [9] proved that the limiting spectral measure of the NTK converges to a deterministic measure, which may be described by recursive fixed-point equations that extend the Marchenko-Pastur distribution.

The present paper is structured as follows. In the first part we consider the non-linear random matrix model studied in [6] and we compute its asymptotic spectral density. We follow the resolvent method via the cumulant expansion which, together with the moment method, is a standard approach to obtain the asymptotics for the empirical spectral density. In particular, we compute the self-consistent equation that is approximately satisfied by the Stieltjes transform of the limiting spectral distribution. This is a quartic equation and is the same as that found in [6]. In [6, 22] the authors relied on the method of moments: they approximated general non-linear functions by polynomial ones and then computed the asymptotics of high moments $\mathbf{E}\,\mathrm{Tr}(Y^k)$ with $Y = f(WX)$ to obtain the limiting measure via its moments. Conversely, we approach matters in a more robust and less combinatorial fashion by applying the resolvent method: we consider $Y$ as a random matrix with correlated entries and then we directly derive a self-consistent equation for its resolvent. In particular, we prove that the random matrix $Y$ has cycle correlations, in the sense that the joint cumulant does not vanish when the random variables $Y_{ij}$'s are joined by a cycle graph. We find that the variance of $Y_{ij}$ is given by the parameter $\theta_1(f)$, whereas for $k > 1$ the cumulants $\kappa(Y_{i_1i_2}, Y^*_{i_2i_3}, Y_{i_3i_4}, \ldots, Y^*_{i_{2k}i_1})$ are powers of $\theta_2(f)$. We note that in the random matrix literature matrices with general decaying correlations have been studied previously, see e.g. [3, 8, 1]. However, the cycle correlations of $Y$ considered in the present paper are much stronger compared to these previous results. The second part of this paper concerns the additive bias case which is a more realistic model for machine learning applications. More precisely, we consider the random feature matrix $Y = f(WX + B)$, where $B$ is a rectangular rank-one Gaussian random matrix, and derive a characterization of the Stieltjes transform of the limiting spectral density. We chose $B$ to be rank-one since for the most commonly used neural network architectures the added bias is chosen equal for each sample. [2] studied the bias case for deterministic data and i.i.d. Gaussian random weights, and computed the exact training error of a ridge-regularized noisy autoenconder in the high-dimensional regime. Interestingly we find that in the case of additive bias it is impossible to choose an activation function $f$ such that the eigenvalue distribution is preserved throughout multiple layers, unlike in the bias-free case where $\theta_2(f) = 0$ yields the Marchenko-Pastur distribution in each layer. Finally, we remark that in the bias-free case our proof via the resolvent method has no significant advantage compared to the moment method, beyond requiring less combinatorics. The main advantage of the resolvent approach is that it allows to include an additive bias without much additional effort.

## 2   Model and main results

We consider a random data matrix $X \in \mathbb{R}^{n_0 \times m}$ with i.i.d. random variables $X_{ij}$ with distribution $\nu_1$ and a random weight matrix $W \in \mathbb{R}^{n_1 \times n_0}$ with i.i.d. weights $W_{ij}$ with distribution $\nu_2$. We assume that both distributions are centred with variance $\mathbf{E}X_{ij}^2 = \sigma_x^2$ and $\mathbf{E}W_{ij}^2 = \sigma_w^2$. Moreover, we assume

that the distributions $\nu_1, \nu_2$ have finite moments of all orders[1]. Since for $1 \leq i \leq n_1$ and $1 \leq j \leq m$ we have

$$\left(\frac{WX}{\sqrt{n_0}}\right)_{ij} = \frac{1}{\sqrt{n_0}} \sum_{k=1}^{n_0} W_{ik} X_{kj},$$

we note that in light of the central limit theorem the entries of the matrix $\frac{WX}{\sqrt{n_0}}$ are approximately $\mathcal{N}(0, \sigma_w^2 \sigma_x^2)$-normally distributed random variables. Therefore, for any $t > 0$, we have the large deviation estimate

$$\mathbf{P}\left(\max_{i,j} \left|\frac{(WX)_{ij}}{\sqrt{n_0}}\right| > t\right) \lesssim n_0^2 \, e^{-t^2/2\sigma_w^2 \sigma_x^2},$$

where we use the notation $A \lesssim B$ as shorthand for the inequality $A \leq cB$ for some constant $c$. Let $f \colon \mathbb{R} \to \mathbb{R}$ be a $C^\infty$ function with zero mean with respect to the Gaussian density of standard deviation $\sigma_w \sigma_x$, i.e.

$$\int_{\mathbb{R}} f(\sigma_w \sigma_x x) \frac{e^{-x^2/2}}{\sqrt{2\pi}} \mathrm{d}x = 0. \tag{2}$$

We consider the random feature model generated by a single-hidden-layer neural network,

$$M = \frac{1}{m} YY^* \in \mathbb{R}^{n_1 \times n_1} \quad \text{with } Y = f\left(\frac{WX}{\sqrt{n_0}}\right), \tag{3}$$

where the activation function $f$ is applied entry-wise. Let $\chi \colon \mathbb{R} \to \mathbb{R}$ be a smooth cut-off function that is equal to one for $|x| \leq 1$ and zero for $|x| \geq 2$. We then replace $f$ by $f(\cdot)\chi(\log^{-1}(n_0)\,\cdot)$. In particular, we now have that $f$ is smooth with compact support. Moreover, for any $l > 0$ and $n_0$ large enough, with probability $1 - n_0^{-l}$, the singular values of $Y$ remain the same.

We are interested in the eigenvalue density of the random matrix $M$ in the infinite size limit. So, we assume that the dimensions of both the columns and the rows of each matrix are large and grow at the same speed, i.e. we introduce some positive constants $\phi$ and $\psi$ such that

$$\frac{n_0}{m} \longrightarrow \phi \quad \text{and} \quad \frac{n_0}{n_1} \longrightarrow \psi \quad \text{as } n_0, n_1, m \to \infty. \tag{4}$$

We denote by $(\lambda_1, \ldots, \lambda_{n_1})$ the eigenvalues of $M$ and define its empirical spectral distribution by $\mu_{n_1} = \frac{1}{n_1} \sum_{i=1}^{n_1} \delta_{\lambda_i}$. Then, as $n_1$ grows large, the empirical distribution of eigenvalues converges in distribution to some deterministic limiting density.

**Theorem 2.1.** *There exists a deterministic measure $\mu = \mu_{\phi,\psi}(\theta_1, \theta_2)$ such that almost surely weakly*

$$\mu_{n_1} \longrightarrow \mu \quad \text{as } n_1 \to \infty.$$

We notice that if $m < n_1$, then $\mathrm{rank}(M) = \min(n_1, m) = m$ and $M$ has $n_1 - m$ zero eigenvalues. In this case, since $\phi/\psi > 1$, there exists an atom at 0 with mass $\mu_{n_1}(0) = 1 - \psi/\phi > 0$, and we have

$$\mu_{n_1} = \frac{n_1 - m}{n_1} \delta_0 + \frac{1}{n_1} \sum_{i=1}^{n_1} \delta_{\lambda_i}.$$

Conversely, if $n_1 < m$, the matrix $M$ has full rank and it is invertible. Since the nonzero eigenvalues of $YY^*$ and $Y^*Y$ are the same, the limiting measure $\mu$ of Theorem 2.1 turns out to be

$$\mu = \left(1 - \frac{\psi}{\phi}\right)_+ \delta_0 + \tilde{\mu},$$

where $(\cdot)_+ = \max(0, \cdot)$, and $\tilde{\mu}$ is the limiting spectral measure of $\frac{1}{m} Y^*Y$.

We will prove that the deterministic measure $\mu$ of Theorem 2.1 is characterized through a quartic self-consistent equation for the Stieltjes transform $g(z) = \frac{1}{n_1} \mathrm{Tr}\, G(z)$ of the empirical spectral measure $\mu_{n_1}$, where

$$G(z) = (M - z)^{-1} \in \mathbb{C}^{n_1 \times n_1}$$

---

[1]This assumption can be relaxed by a customary cut-off argument, but we refrain from doing so for simplicity.

is the resolvent of the random matrix $M$ and the spectral parameter $z$ lies in the upper half plane $\mathbb{H} = \{z \in \mathbb{C} \,|\, \Im z \geq 0\}$. We set

$$\theta_1(f) := \int_{\mathbb{R}} f^2(\sigma_w \sigma_x x) \frac{e^{-x^2/2}}{\sqrt{2\pi}} \mathrm{d}x \quad \text{and} \quad \theta_2(f) := \left( \sigma_w \sigma_x \int_{\mathbb{R}} f'(\sigma_w \sigma_x x) \frac{e^{-x^2/2}}{\sqrt{2\pi}} \mathrm{d}x \right)^2. \tag{5}$$

Then, the following theorem characterizes $g$ as the solution to a quartic equation which depends only on the two parameters $\theta_1(f)$ and $\theta_2(f)$.

**Theorem 2.2.** *For some $\delta, \epsilon > 0$ and any $z \in \mathbb{H}$ with $\Im z > n_1^{-\frac{1}{4}+\epsilon}$, the measure $\mu$ is characterized through the following self-consistent equation*

$$\left| 1 + zg - \left( \theta_1 - \frac{\theta_2}{\psi}(1 + zg) \right) g \left( 1 - \frac{\phi}{\psi}(1 + zg) \right) - \frac{\theta_2(\theta_1 - \theta_2)}{\psi} g^2 \left( 1 - \frac{\phi}{\psi}(1 + zg) \right)^2 \right| \leq n_1^{-\delta} \tag{6}$$

*almost surely.*

**Remark 2.3.**

*(i) We obtain an analogous result for complex feature and weight matrices $W, X$, c.f. Remark E.1.*

*(ii) Note that the quartic self-consistent equation (6) may not have a unique solution such that $\Im g(z) > 0$. However, it has a unique solution which is analytic in the upper half-plane and satisfies $g(z) \sim -1/z$ for large $|z|$.*

*(iii) Since the resolvent itself satisfies $\operatorname{Tr} G(z)/n_1 \sim -1/z$ for large $|z|$ and is analytic in the upper half-plane, by continuity Theorem 2.2 implies that $g(z)$ is approximately given by a properly chosen solution of (6). Then, the limiting spectral measure itself can be recovered via the Stieltjes inversion formula,*

$$\mu(\lambda) = \lim_{\epsilon \to 0^+} \frac{1}{\pi} \Im g(\lambda + i\epsilon),$$

*and Theorem 2.1 follows from Theorem 2.2.*

*(iv) It follows from the self-consistent equation (6) that the limiting spectral measure $\tilde{\mu}$ is absolutely continuous w.r.t. the Lebesgue measure, and therefore so is $\mu$ away from zero. Moreover, for large $z$, equation (6) has real solutions and thus via Stieltjes inversion the limiting measure $\mu$ is compactly supported.*

**Remark 2.4.** *It should be noted that Theorem 2.1 and Theorem 2.2 were proven in [22, 6] under different assumptions and with a different method. The result in [22] was obtained for i.i.d. Gaussian features and weights, whereas [6] extends the result to the case where both the inputs and the random weights have sub-Gaussian tails but are not necessarily Gaussian.*

Observing equation (6), we note that if $\theta_2(f) = 0$, then the limiting measure $\mu$ is exactly the Marchenko-Pastur $\mu_{MP}$ distribution with parameter $\phi/\psi$. Indeed, in this case, $g(z)$ approximately satisfies the quadratic equation

$$1 + \left( z + \theta_1(f) \left( \frac{\phi}{\psi} - 1 \right) \right) g(z) + \theta_1(f) \frac{\phi}{\psi} z g(z)^2 \approx 0, \tag{7}$$

which corresponds to the self-consistent equation satisfied by the Stieltjes transform of $\mu_{MP}$ [20]. As discussed in the introduction, this consideration is relevant when studying multilayer networks. Pennington and Worah [22] conjectured that the asymptotic spectral distribution is preserved through multiple layers only by activation functions with $\theta_2(f) = 0$ and is given by the Marchenko-Pastur distribution in each layer. Benigni and Péché [6] then proved this conjecture for bounded activation functions satisfying $\theta_2(f) = 0$. Moreover, if $\theta_1(f) = \theta_2(f)$, then equation (6) becomes cubic. In particular, the equality $\theta_1(f) = \theta_2(f)$ holds if and only if $f$ is a linear function (for more details, we refer to the supplementary material in [22]). In this case, $M = \frac{1}{m} YY^*$ with $Y = WX$, and thus the limiting measure $\mu$ corresponds to the limiting spectral distribution of a product Wishart matrix. The spectral density for matrices of this type has been computed in [7, 10].

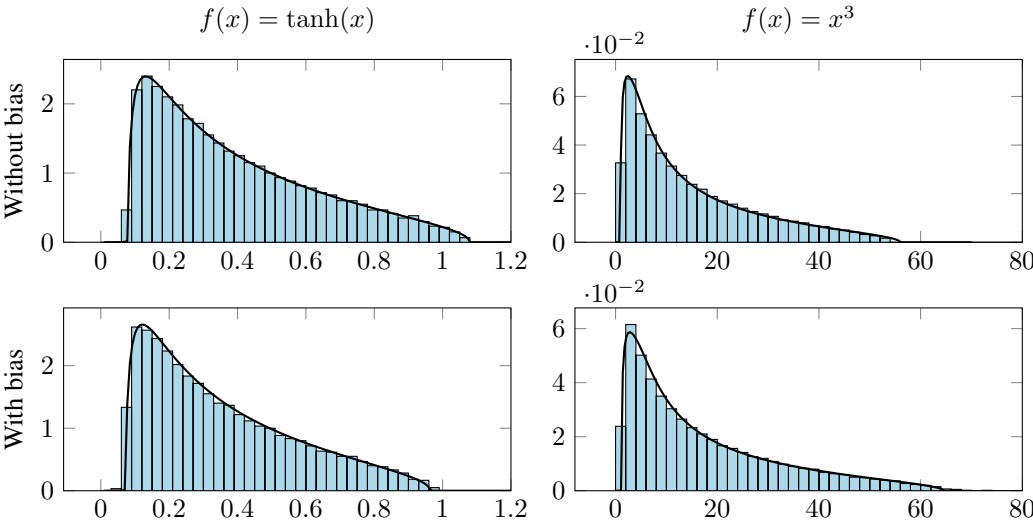

Figure 1: We present the eigenvalue histogram of the covariance matrix $YY^*$ for a single random realisation together with the theoretical limit from Theorems 2.1 and 2.5 for the functions $f(x) = \tanh(x)$ and $f(x) = x^3$ with and without additive bias. We note that the presence of an additive bias can both increase or decrease the largest singular value. The numerical experiments were conducted for the parameters $n_1 = 3000$, $\phi = \sigma_x = \sigma_w = 1$, $\psi = 5$ (left) or $\psi = 2$ (right), and $\sigma_b = 0$ (top) or $\sigma_b = 0.25$ (bottom).

## 2.1 Additive bias case

The previous model can be generalised by adding random biases. In neural networks, the bias is an additional parameter that allows the model to better fit the given data. In this case, for each input data $x \in \mathbb{R}^{n_0}$, a bias vector $b \in \mathbb{R}^{n_1}$ is added to the vector $Wx \in \mathbb{R}^{n_1}$. We then apply a non-linear function $f \colon \mathbb{R} \to \mathbb{R}$ in an element-wise fashion to its vector arguments $Wx + b$ in order to obtain $n_1$ random features $f(Wx + b) \in \mathbb{R}^{n_1}$.

We consider a random bias matrix $B \in \mathbb{R}^{n_1 \times m}$ of i.i.d. Gaussian random variables $B_{ij} = B_i$ with zero mean and variance $\mathbf{E}B_i^2 = \sigma_b^2$. Note that the random matrix $B$ has rank 1. Let $X \in \mathbb{R}^{n_0 \times m}$ and $W \in \mathbb{R}^{n_1 \times n_0}$ be random matrices with i.i.d. entries, defined as before. Moreover, let $f \colon \mathbb{R} \to \mathbb{R}$ be a $C^\infty$ function satisfying

$$\int_{\mathbb{R}} f\left(\sqrt{\sigma_w^2 \sigma_x^2 + \sigma_b^2}\, x\right) \frac{e^{-x^2/2}}{\sqrt{2\pi}}\mathrm{d}x = 0. \tag{8}$$

Just as before, without loss of generality, upon replacing $f$ by $f(\cdot)\chi(\log^{-1}(n_0)\cdot)$, we may assume that $f$ is a smooth function with compact support. We then define the random matrix $M$ by

$$M = \frac{1}{m}YY^* \in \mathbb{R}^{n_1 \times n_1} \quad \text{with } Y = f\left(\frac{WX}{\sqrt{n_0}} + B\right), \tag{9}$$

where $f$ is applied entry-wise. We introduce the parameter

$$\widetilde{\sigma} := \sqrt{\frac{\sigma_w^2 \sigma_x^2 (\sigma_w^2 \sigma_x^2 + 2\sigma_b^2)}{\sigma_w^2 \sigma_x^2 + \sigma_b^2}},$$

and we define the following integral parameters:

$$\theta_1(f) := \int_{\mathbb{R}} f^2\left(\sqrt{\sigma_w^2\sigma_x^2 + \sigma_b^2}\,x\right)\frac{e^{-x^2/2}}{\sqrt{2\pi}}\,\mathrm{d}x,$$

$$\theta_{1,b}(f) := \frac{1}{2\pi\widetilde{\sigma}\sqrt{\sigma_w^2\sigma_x^2 + \sigma_b^2}}\int_{\mathbb{R}^2} f(x_1)f(x_2)\exp\left(-\frac{x_1^2 + x_2^2}{2\widetilde{\sigma}^2} + \frac{\sigma_b^2 x_1 x_2}{\widetilde{\sigma}^2(\sigma_w^2\sigma_x^2 + \sigma_b^2)}\right)\mathrm{d}\boldsymbol{x}, \quad (10)$$

$$\theta_2(f) := \frac{\sigma_w\sigma_x}{2\pi\widetilde{\sigma}\sqrt{\sigma_w^2\sigma_x^2 + \sigma_b^2}}\int_{\mathbb{R}^2} f'(x_1)f'(x_2)\exp\left(-\frac{x_1^2 + x_2^2}{2\widetilde{\sigma}^2} + \frac{\sigma_b^2 x_1 x_2}{\widetilde{\sigma}^2(\sigma_w^2\sigma_x^2 + \sigma_b^2)}\right)\mathrm{d}\boldsymbol{x}.$$

We can now state the analogue of Theorem 2.2 in the additive bias case. In particular, the following theorem shows that the normalized trace of the resolvent of $M$ approximately satisfies the self-consistent equation (6) with parameters given by (10).

**Theorem 2.5.** *The Stieltjes transform g satisfies* (6) *with parameters given by* (10)*, where $\theta_1(f)$ is replaced by $\theta_1(f) - \theta_{1,b}(f)$. Moreover, there exists a single outlier eigenvalue $\lambda_{\max} = n_1\theta_{1,b}(1 + \mathcal{O}(n_1^{-1/2}))$ of $M$ that is separated from the support of the rest of the spectrum.*

We remark that the parameters $\theta_{1,b}(f), \theta_2(f)$ can be alternatively expressed as infinite series, directly demonstrating that for $\sigma_b \neq 0$ and non-trivial $f$ both coefficients are strictly positive, $\theta_{1,b}(f), \theta_2(f) > 0$. For notational implicitly, we introduce the Hermite inner product

$$\langle f, g \rangle_{\mathrm{He}} := \frac{1}{\sqrt{2\pi}}\int_{\mathbb{R}} f(x)g(x)e^{-x^2/2}\,\mathrm{d}x.$$

**Remark 2.6.** *We have*

$$\theta_{1,b}(f) = \frac{\widetilde{\sigma}}{\sqrt{\sigma_w^2\sigma_x^2 + \sigma_b^2}}\sum_{k \geq 0}\frac{1}{k!}\left(\frac{\sigma_b^2}{\sigma_w^2\sigma_x^2 + \sigma_b^2}\right)^k\langle x^k, f(\widetilde{\sigma}\cdot)\rangle_{\mathrm{He}}^2$$

$$\theta_2(f) = \frac{\sigma_w^2\sigma_x^2\widetilde{\sigma}}{\sqrt{\sigma_w^2\sigma_x^2 + \sigma_b^2}}\sum_{k \geq 0}\frac{1}{k!}\left(\frac{\sigma_b^2}{\sigma_w^2\sigma_x^2 + \sigma_b^2}\right)^k\langle x^k, f'(\widetilde{\sigma}\cdot)\rangle_{\mathrm{He}}^2 \tag{11}$$

*and therefore $\theta_{1,b}(f) = 0$, $\sigma_b \neq 0$ implies that $f(\widetilde{\sigma}\cdot)$ is orthogonal to Hermite polynomials of any order, and consequently $f \equiv 0$. Similarly, $\theta_2(f) = 0$, $\sigma_b \neq 0$ implies that $f \equiv$ const.*

## 2.2 Multiple layers

In [22] it was observed empirically that in the bias-free case activation functions with $\theta_2(f) = 0$ have the remarkable property that for multiple layers

$$Y^{(l+1)} := f(W^{(l)}Y^{(l)}), \qquad Y^{(0)} := X \tag{12}$$

the singular value distributions of $Y^{(1)}, Y^{(2)}, \dots$ all asymptotically agree (up to scaling) with the probability distribution $\mu(\theta_1, \theta_2) = \mu(\theta_1, 0)$ from Theorem 2.1. This observation is very natural from our point of view since we find that $Y^{(1)}$ is approximately an i.i.d. random matrix if $\theta_2(f) = 0$, c.f. Proposition 3.2 below.

An interesting corollary of our Theorem 2.5 is that a similar isospectral property *cannot* be ensured for the case of additive bias

$$Y^{(l+1)} := f(W^{(l)}Y^{(l)} + B^{(l)}), \qquad Y^{(0)} := X. \tag{13}$$

Indeed, in light of Remark 2.6, for $\sigma_b \neq 0$ we have $\theta_{1,b}(f), \theta_2(f) > 0$ for all activation functions $f$, and therefore already the random matrix $Y^{(1)}$ necessarily has leading order correlations, c.f. Proposition 3.3 below. Hence, convergence of the spectral density to the solution of (6) is not expected beyond the first layer. In Fig. 2 we test this result experimentally and choose the activation function $f(x) = c_1|x| - c_2$ with $c_1, c_2$ such that (2) is satisfied and $\theta_1(f) = 1$. We find that in the bias-free case (left), irrespective of the network depth, the eigenvalues of the covariance matrix $Y^{(l)}(Y^{(l)})^*$ converge to their theoretical limit from Theorem 2.1, exactly as in [22, Fig. 1][2]. In the case of an additive bias (right), no such convergence is observed, and this provides empirical evidence of our result.

---

[2]In the notation of [22], $f = f_1$.

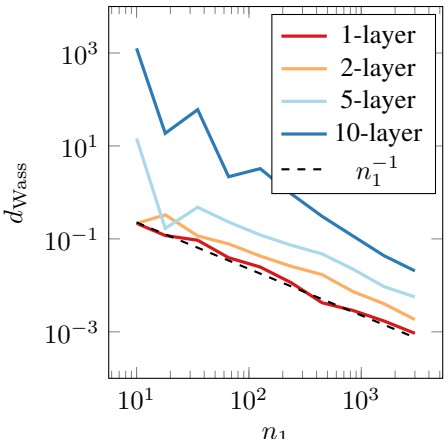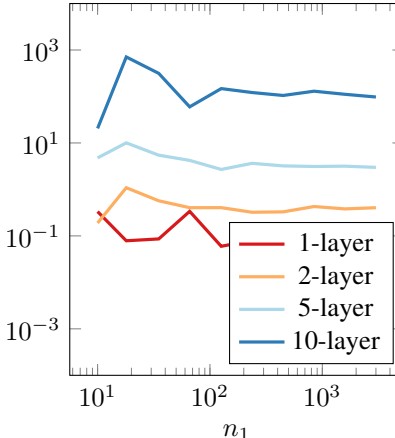

Figure 2: For randomly generated neural networks of varying depth and width, we compute the Wasserstein distance $d_{\text{Wass}}$ between the empirical eigenvalue density of the covariance matrix $Y^{(l)}(Y^{(l)})^*$ to the distribution $\mu$ from Theorem 2.1 for the activation function $f(x) = c_1|x| - c_2$. In the bias-free case (left), the Wasserstein distance decays as the inverse of the network width, while in the case of an additive bias (right) no convergence can be observed. The numerical experiments were conducted for the parameters $\phi = \sigma_x = \sigma_w = 1$, $\psi = 2$ and $\sigma_b = 0$ (left) or $\sigma_b = 0.5$ (right).

The spectrum of the covariance matrix $Y^{(l)}(Y^{(l)})^*$ reflects the distortion of input data through the network and highly skewed distributions indicate poor conditioning which may impede learning performance [22]. *Batch normalization* seeks to remedy the distortion by normalising by the trace of the covariance matrix $Y^{(l)}(Y^{(l)})^t$ in each layer. In [22] it was suggested that choosing activation functions with $\theta_2(f) = 0$, i.e. functions which naturally preserve the singular value distribution, may serve as an alternative method of tuning networks for fast optimisation. Our result indicates that in the case of additive bias this alternative is not present. However, batch normalization seems to help stabilising the singular value distribution also in the additive bias case, c.f. Fig. 3.

## 3 Outline of proof of Theorems 2.2 and 2.5

The proof of both Theorem 2.2 and 2.5 can be broken into two distinct parts. The first step is to show that $Y = f\left(\frac{WX}{\sqrt{n_0}}\right) \in \mathbb{R}^{n_1 \times m}$ can be viewed as a correlated random matrix with *cycle correlations*, c.f. Propositions 3.2 and 3.3 below. The second step is to prove the global law for the random matrix $M = \frac{1}{m}YY^*$ with the cycle correlations. In the following, we will sketch the derivation of the self-consistent equation. A more detailed proof is provided in the supplementary material.

The key idea is to use a multivariate cumulant expansion formula. Cumulants of a random vector $\boldsymbol{X} = (X_1, \dots, X_n)$ can be defined in a combinatorial way by

$$\kappa(X_1, \dots, X_n) = \sum_{\pi} (-1)^{|\pi|-1}(|\pi| - 1)! \prod_{B \in \pi} \mathbf{E}\left(\prod_{i \in B} X_i\right), \qquad (14)$$

where the sum runs over all partitions $\pi$ of the set $[n] = \{1, \dots, n\}$, the product runs over the blocks $B$ of the partition $\pi$, and $|\pi|$ is the number of blocks in the partition. The following expansion is commonly referred to as a cumulant expansion and generalises the Gaussian integration by parts. In the context of random matrix theory, the usefulness of this expansion was first observed in [15] and later revived in [12, 13]. A proof of the following lemma is provided in Appendix C for completeness.

**Lemma 3.1** (Cumulant expansion). *If $\boldsymbol{X} = (X_1, \dots, X_n)$ is a random vector with finite moments of all orders, then*

$$\mathbf{E}X_1 f(\boldsymbol{X}) = \sum_{l \geq 1} \sum_{i_1, \dots, i_l} \frac{\kappa(X_1, X_{i_1}, \dots, X_{i_l})}{l!} \mathbf{E}\partial_{i_1} \cdots \partial_{i_l} f(\boldsymbol{X}),$$

*where $f \colon \mathbb{R}^n \to \mathbb{R}$ is smooth.*

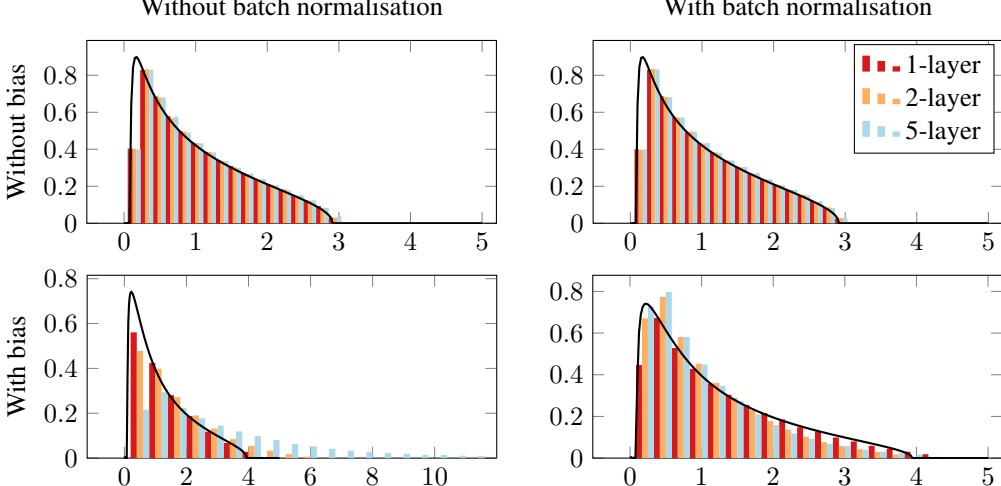

Figure 3: We present the eigenvalue distribution of neural networks of varying depth and in the presence/absence of both bias and batch normalization for the activation function $f(x) = c_1|x| - c_2$. In the bias-free case, batch normalisation has no effect on the spectral stability, and throughout the network the theoretical distribution from Theorem 2.1 matches the actual eigenvalue distribution of the covariance matrix $Y^{(l)}(Y^{(l)})^*$ well. In the case of an additive bias, the single-layer spectral density matches the theoretical limit from Theorem 2.5 to high accuracy. However, for multiple layers the spectral density diverges without additional batch normalization. Batch normalization alleviates the divergence, however the actual eigenvalue distribution deviates from the theoretical limit from Theorem 2.5. The numerical experiments were conducted for the parameters $n_1 = 3000$, $\phi = \sigma_x = \sigma_w = 1$, $\psi = 2$ and $\sigma_b = 0$ (top) or $\sigma_b = 0.5$ (bottom). Here we used batch normalisation of the form $Y^{(l)} \mapsto cY^{(l)}$ after each layer, choosing $c$ to ensure unit empirical variance.

We start with the defining identity of the resolvent, $\mathbf{1}_{n_1} + zG = MG$, where $\mathbf{1}_{n_1}$ denotes the $n_1 \times n_1$ identity matrix, and we compute its average trace:

$$1 + zg = \frac{1}{n_1} \operatorname{Tr} \frac{YY^*G}{m} = \frac{1}{n_1} \sum_{i=1}^{n_1} \sum_{j=1}^{m} Y_{ij} \left( \frac{Y^*G}{m} \right)_{ji}, \qquad (15)$$

where $g(z) = \frac{1}{n_1} \operatorname{Tr}(M - z\mathbf{1}_{n_1})^{-1}$ is the normalized trace of the resolvent of $M$. Since the random variable $(Y^*G)_{ji}$ can be seen as a function of $Y_{ij}$, we can take the expectation on both sides of (15) and apply Lemma 3.1:

$$1 + z\mathbf{E}g = \frac{1}{n_1} \sum_{k \geq 1} \sum_{i_1,\ldots,i_{2k}} \frac{\kappa(Y_{i_1 i_2}, Y_{i_3 i_4}, \ldots, Y_{i_{2k-1} i_{2k}})}{(k-1)!} \mathbf{E} \partial_{Y_{i_3 i_4}} \cdots \partial_{Y_{i_{2k-1} i_{2k}}} \left( \frac{Y^*G}{m} \right)_{i_2 i_1}. \quad (16)$$

The main goal now is to show that $Y$ can be viewed as a random matrix with cycle correlations given as in the Propositions 3.2 and 3.3 below: Prop. 3.2 refers to the bias-free case and Prop. 3.3 to the additive bias case. We postpone the proof of both propositions to Subsections A.2 and B.2, resp.

**Proposition 3.2** (Correlation structure without bias). *The random matrix $Y$ defined by (3) has joint cumulants given by*

$$\kappa(Y_{i_1 i_2}) = \mathcal{O}(n_0^{-1/2}),$$
$$\kappa(Y_{i_1 i_2}, Y_{i_2 i_1}^*) \approx \theta_1(f), \qquad (17)$$
$$\kappa(Y_{i_1 i_2}, Y_{i_2 i_3}^*, Y_{i_3 i_4}, \ldots, Y_{i_{2k} i_1}^*) \approx \theta_2(f)^k n_0^{1-k}, \quad k > 1$$

*where $i_1, \ldots, i_{2k}$ are all distinct, and we write $X \approx Y$ as a shorthand notation for $X = Y(1 + \mathcal{O}(n_0^{-1/2}))$.*

**Proposition 3.3** (Correlation structure with bias). *The random matrix $Y$ defined by* (9) *has joint cumulants given by*

$$\begin{aligned}
\kappa(Y_{i_1 i_2}) &= \mathcal{O}(n_0^{-1/2}), \\
\kappa(Y_{i_1 i_2}, Y_{i_2 i_1}^*) &\approx \theta_1(f), \\
\kappa(Y_{i_1 i_2}, Y_{i_3 i_1}^*) &\approx \theta_{1,b}(f) \\
\kappa(Y_{i_1 i_2}, Y_{i_2 i_3}^*, Y_{i_3 i_4}, \dots, Y_{i_{2k} i_1}^*) &\approx \theta_2(f)^k n_0^{1-k}, \quad k > 1
\end{aligned}$$

(18)

*where $i_1, \dots, i_{2k}$ are all distinct.*

Applying Propositions 3.2 and 3.3 to (16), computing the partial derivatives and doing some book-keeping, we get the desired equation (6) as $n_0, n_1, m \to \infty$. To complete the proofs of Theorems 2.2 and 2.5, one has to show the concentration of $g$ around $\mathbf{E}g$, as stated in the following lemma.

**Lemma 3.4.** *For the random matrix $M = \frac{1}{m} Y Y^*$ and a complex number $z \in \mathbb{H}$ such that $\Im z > n_1^{-\frac{1}{4}+\epsilon}$, for some $\epsilon > 0$, it holds that*

$$\mathbf{E}_W |g(z) - \mathbf{E}_W g(z)|^4 \lesssim \frac{1}{n_1^2 (\Im z)^4}$$

(19)

*with high probability in $X$, and analogously*

$$\mathbf{E}_X |g(z) - \mathbf{E}_X g(z)|^4 \lesssim \frac{1}{n_1^2 (\Im z)^4}$$

(20)

*with high probability in $W$, where $\mathbf{E}_X$ (resp. $\mathbf{E}_W$) is the expectation in the $X$-space (resp. $W$-space).*

The proof of this lemma relies on a standard argument (e.g. see the proof of the concentration inequality in [5, Subsection 3.3.2]) and is given in Appendix D.

# 4 Conclusion

In this paper, we analysed the singular value distribution of fully random neural networks and found that in the case of additive biases it is impossible to achieve isospectrality by tuning the activation function. In addition, we showed that the resolvent method from random matrix theory also applies to the neural network analysis, despite the non-linearities and we expect that this robust method will prove to be useful in contexts where the conventionally used moment method becomes intractable.

## Broader impact

Our result is a purely theoretical one for fully random features, weights and biases. Therefore, we do not expect our contribution to have ethical concerns or adverse future societal consequences.

## Acknowledgments and Disclosure of Funding

D. Schröder would like to thank L. Benigni for illuminating discussions on the subject and both authors would like to thank him for his helpful comments on an early version of this preprint. Both authors thank the referees for their careful reading of our manuscript. This work was carried out when the first author was a research assistant at ETH Zurich in the group of W. Werner. The second author is supported by Dr. Max Rössler, the Walter Haefner Foundation and the ETH Zürich Foundation.

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
