# A  Proof of Theorem 2.2

## A.1  Derivation of the self-consistent equation

We start from (16) and rely on the following power counting principles: Each derivative provides a smallness-factor of $1/\sqrt{m}$ because $G$ is a function of $Y/\sqrt{m}$ and $Y^*/\sqrt{m}$, while each independent summation costs a factor of $n_1 \sim m$. However, we cannot have too many independent summations for if any index appears only once in the cumulant, then the latter vanishes identically by the independence property of cumulants. For example, if $i_2, \ldots, i_{2k} \neq i_1$, then the random variables $Y_{i_3 i_4}, \ldots, Y_{i_{2k-1} i_{2k}}$ are independent of $Y_{i_1 i_2}$ in the probability space of the random variables $\left\{ w_{i_1 a} \right\}_{a=1}^{n_0}$ conditioned on the remaining random variables. By the law of total expectation and the independence property it follows that

$$\kappa(Y_{i_1 i_2}, \ldots, Y_{i_{2k-1} i_{2k}}) = 0$$

in this case. Thus we only need to sum over those cumulants in which each $W$- and $X$-index appears at least twice (we call $i$ the $W$-index of $Y_{ij}, Y_{ji}^*$ and $j$ the $X$-index). In the extreme case where each $W$- and $X$-index appears exactly twice, we either have a single cycle, or a union of cycles on disjoint index sets. In the latter case the cumulant vanishes identically by the independence property. In the former case, for a cycle of length $2k$ there are $k$ indices each, we obtain a factor of $n_1^{-1}$ from the normalised sum, a factor of $m^{-2k/2} = m^{-k}$ from the derivatives, a factor of $n_1^k m^k$ from the summations, and finally a factor of $n_0^{1-k}$ from the cumulant in Proposition 3.2, i.e.

$$\frac{1}{n_1} \frac{1}{m^k} n_1^k m^k n_0^{1-k} \sim 1$$

and the power counting is neutral. On the contrary, when some index appears three times, the overall power counting described above is smaller by a factor of $1/\sqrt{m}$, and thus negligible to leading order. In particular this argument shows that cycles of odd length only negligible as they cannot arise on indices in which each $W$- and $X$-index appears exactly twice.

Thus, together with Proposition 3.2 we have (recalling that the shorthand notation $\approx$ indicates equalities up to an error of $n_0^{-1/2}$)

$$
\begin{aligned}
1 + z\,\mathbf{E}g &= \frac{1}{n_1 m} \sum_{k \geq 1} \sum_{i_1, \ldots, i_{2k}} \frac{\kappa(Y_{i_1 i_2}, Y_{i_3 i_4}, Y_{i_5 i_6}, \ldots, Y_{i_{2k-1} i_{2k}})}{(k-1)!} \,\mathbf{E}\partial_{Y_{i_3 i_4}} \cdots \partial_{Y_{i_{2k-1} i_{2k}}} (Y^* G)_{i_2 i_1} \\
&\approx \frac{1}{n_1 m} \sum_{k \geq 1} \sum_{i_1, \ldots, i_{2k}}^{*} \kappa(Y_{i_1 i_2}, Y_{i_2 i_3}^*, Y_{i_3 i_4}, \ldots, Y_{i_{2k} i_1}^*) \,\mathbf{E}\partial_{Y_{i_3 i_4}} \cdots \partial_{Y_{i_{2k-1} i_{2k}}} (Y^* G)_{i_2 i_1} \\
&= \frac{1}{n_1 m} \sum_{i_1, i_2}^{*} \kappa(Y_{i_1 i_2}, Y_{i_2 i_1}^*) \,\mathbf{E}\partial_{Y_{i_2 i_1}^*} (Y^* G)_{i_2 i_1} \\
&\quad + \frac{1}{n_1 m} \sum_{k \geq 2} \sum_{i_1, \ldots, i_{2k}}^{*} \kappa(Y_{i_1 i_2}, Y_{i_2 i_3}^*, Y_{i_3 i_4}, \ldots, Y_{i_{2k} i_1}^*) \,\mathbf{E}\partial_{Y_{i_2 i_3}^*} \cdots \partial_{Y_{i_{2k} i_1}^*} (Y^* G)_{i_2 i_1} \\
&\approx \frac{\theta_1}{n_1 m} \sum_{i_1, i_2}^{*} \mathbf{E}\partial_{Y_{i_2 i_1}^*} (Y^* G)_{i_2 i_1} + \frac{1}{n_1 m} \sum_{k \geq 2} \frac{\theta_2^k}{n_0^{k-1}} \sum_{i_1, \ldots, i_{2k}}^{*} \mathbf{E}\partial_{Y_{i_2 i_3}^*} \cdots \partial_{Y_{i_{2k} i_1}^*} (Y^* G)_{i_2 i_1} ,
\end{aligned}
$$

$$(21)$$

where the summations $\sum^{*}$ are understood over pairwise distinct indices. Here in the second line the factorial $(k-1)!$ disappears since there are exactly $(k-1)!$ ways to map the variables $Y_{i_3 i_4}, Y_{i_5 i_6} \ldots, Y_{i_{2k-1} i_{2k}}$ into $Y_{i_2 i_3}^*, Y_{i_3 i_4}, \ldots, Y_{i_{2k} i_1}^*$ with distinct $i_1, \ldots, i_{2k}$. From this point onwards, we will omit reference to $\mathbf{E}$ to simplify notation slightly.

We now need to compute the partial derivatives in (21). The proof of the following lemma is included in Appendix C.

**Lemma A.1.** *Let $G(z) = (M - z)^{-1}$, $z \in \mathbb{H}$, be the resolvent of the random matrix $M = \frac{1}{m}YY^* \in \mathbb{R}^{n_1 \times n_1}$. Then, it holds that*

$$\partial_{Y^*_{i_2 i_1}} (Y^* G)_{i_2 i_1} = G_{i_1 i_1} \left( 1 - \left( \frac{Y^* G Y}{m} \right)_{i_2 i_2} \right), \tag{22a}$$

$$\partial_{Y^*_{i_2 i_3}} \cdots \partial_{Y^*_{i_{2k} i_1}} (Y^* G)_{i_2 i_1} \approx -\partial_{Y_{i_3 i_4}} \cdots \partial_{Y_{i_{2k-1} i_{2k}}} \left( \frac{GY}{m} \right)_{i_3 i_{2k}} G_{i_1 i_1} \left( 1 - \left( \frac{Y^* G Y}{m} \right)_{i_2 i_2} \right). \tag{22b}$$

Thus, using Lemma A.1 in (21) we have

$$1 + zg \approx \frac{\theta_1}{n_1 m} \sum_{i_1, i_2}^* G_{i_1 i_1} \left( 1 - \left( \frac{Y^* G Y}{m} \right)_{i_2 i_2} \right)$$

$$- \frac{1}{n_1 m} \sum_{k \geq 2} \frac{\theta_2^k}{n_0^{k-1}} \sum_{i_1, \ldots, i_{2k}}^* \partial_{Y_{i_3 i_4}} \cdots \partial_{Y_{i_{2k-1} i_{2k}}} \left( \frac{GY}{m} \right)_{i_3 i_{2k}} G_{i_1 i_1} \left( 1 - \left( \frac{Y^* G Y}{m} \right)_{i_2 i_2} \right)$$

$$= \theta_1 g - \theta_1 \frac{n_1}{m} g \left\langle \frac{Y^* G Y}{m} \right\rangle$$

$$- \left( g - \frac{n_1}{m} g \left\langle \frac{Y^* G Y}{m} \right\rangle \right) \frac{1}{m} \sum_{k \geq 2} \frac{\theta_2^k}{n_0^{k-1}} \sum_{i_3, \ldots, i_{2k}}^* \partial_{Y_{i_3 i_4}} \cdots \partial_{Y_{i_{2k-1} i_{2k}}} (GY)_{i_3 i_{2k}}, \tag{23}$$

where $\left\langle \frac{Y^* G Y}{m} \right\rangle := \frac{1}{n_1} \operatorname{Tr} \frac{Y^* G Y}{m} = 1 + zg$ from (15). Again, we stress that the equalities are meant in expectation. Moreover, shifting the index in the above summation, we get

$$\frac{1}{m} \sum_{k \geq 2} \frac{\theta_2^k}{n_0^{k-1}} \sum_{i_3, \ldots, i_{2k}}^* \partial_{Y_{i_3 i_4}} \cdots \partial_{Y_{i_{2k-1} i_{2k}}} (GY)_{i_3 i_{2k}}$$

$$= \theta_2 \frac{n_1}{n_0} \frac{1}{m} \sum_{k \geq 1} \frac{\theta_2^k}{n_1 n_0^{k-1}} \sum_{i_3, \ldots, i_{2k+2}}^* \partial_{Y_{i_3 i_4}} \cdots \partial_{Y_{i_{2k+1} i_{2k+2}}} (GY)_{i_3 i_{2k+2}}$$

$$= \theta_2^2 \frac{n_1}{n_0} \frac{1}{n_1 m} \sum_{i_3, i_4}^* \partial_{Y_{i_3 i_4}} (GY)_{i_3 i_4}$$

$$+ \theta_2 \frac{n_1}{n_0} \frac{1}{n_1 m} \sum_{k \geq 2} \frac{\theta_2^k}{n_0^{k-1}} \sum_{i_3, \ldots, i_{2k+2}}^* \partial_{Y_{i_3 i_4}} \cdots \partial_{Y_{i_{2k+1} i_{2k+2}}} (GY)_{i_3 i_{2k+2}}$$

$$\approx \theta_2^2 \frac{n_1}{n_0} \left( g - \frac{n_1}{m} g \left\langle \frac{Y^* G Y}{m} \right\rangle \right) + \theta_2 \frac{n_1}{n_0} \left( 1 + zg - \theta_1 g + \theta_1 \frac{n_1}{m} g \left\langle \frac{Y^* G Y}{m} \right\rangle \right)$$

$$= \theta_2 \frac{n_1}{n_0} (1 + zg) - \theta_2 (\theta_1 - \theta_2) \frac{n_1}{n_0} g \left( 1 - \frac{n_1}{m} (1 + zg) \right),$$

where in the third step we used (21). Finally, together with (23), we have

$$1 + zg \approx \theta_1 g \left( 1 - \frac{n_1}{m} (1 + zg) \right) - \theta_2 \frac{n_1}{n_0} g (1 + zg) \left( 1 - \frac{n_1}{m} (1 + zg) \right)$$

$$+ \theta_2 (\theta_1 - \theta_2) \frac{n_1}{n_0} g^2 \left( 1 - \frac{n_1}{m} (1 + zg) \right)^2, \tag{24}$$

which corresponds to the desired equation (6) as $n_0, n_1, m \to \infty$. Thus, (24) combined with the concentration inequality given in Lemma 3.4 completes the proof of Theorem 2.2.

*Proof of Theorem 2.2.* We need to show the concentration w.r.t. $\mathbf{E}_{W,X} \equiv \mathbf{E}$. By the triangle and Jensen inequality we have

$$\mathbf{E}|g(z) - \mathbf{E}g(z)|^4 \lesssim \mathbf{E}|g(z) - \mathbf{E}_W g(z)|^4 + \mathbf{E}_X |\mathbf{E}_W g(z) - \mathbf{E}g(z)|^4$$

$$\leq \mathbf{E}_X \left( \mathbf{E}_W |g(z) - \mathbf{E}_W g(z)|^4 \right) + \mathbf{E}_W \left( \mathbf{E}_X |g(z) - \mathbf{E}_X g(z)|^4 \right) \lesssim \frac{2}{n_1^2 (\Im z)^4}$$

and thus the almost sure convergence follows from the Borel-Cantelli Lemma, completing the proof of Theorem 2.2 together with (24). □

## A.2 Proof of Proposition 3.2

In light of the central limit theorem, we have that in the asymptotic limit the random variables

$$\left(\frac{WX}{\sqrt{n_0}}\right)_{ij} = \frac{1}{\sqrt{n_0}}\sum_{k=1}^{n_0} W_{ik}X_{kj},$$

are approximately $\mathcal{N}(0, \sigma_w^2\sigma_x^2)$-normally distributed. Our next goal is to compute their cumulants. The first cumulant or expectation vanishes identically. For the second cumulant we obtain:

**Lemma A.2.** *The cumulant of* $\frac{(WX)_{i_1i_2}}{\sqrt{n_0}}$ *and* $\frac{(WX)_{i_3i_4}}{\sqrt{n_0}}$ *is nonzero only if* $i_1 = i_3$ *and* $i_2 = i_4$, *and in this case it holds that*

$$\kappa\left(\frac{(WX)_{i_1i_2}}{\sqrt{n_0}}, \frac{(WX)^*_{i_2i_1}}{\sqrt{n_0}}\right) = \sigma_w^2\sigma_x^2.$$

*Proof.* We have

$$\kappa\left(\frac{(WX)_{i_1i_2}}{\sqrt{n_0}}, \frac{(WX)_{i_3i_4}}{\sqrt{n_0}}\right) = \frac{1}{n_0}\mathbf{E}(WX)_{i_1i_2}(WX)_{i_3i_4}$$

$$= \frac{1}{n_0}\sum_{k_1,k_2=1}^{n_0}\mathbf{E}W_{i_1k_1}X_{k_1i_2}W_{i_3k_2}X_{k_2i_4}$$

$$= \frac{1}{n_0}\sum_{k_1=1}^{n_0}\delta_{i_1i_3}\delta_{i_2i_4}\,\mathbf{E}W_{i_1k_1}^2X_{k_1i_2}^2 = \delta_{i_1i_3}\delta_{i_2i_4}\sigma_w^2\sigma_x^2.$$

Thus, the second cumulant is nonzero if $i_1 = i_3$ and $i_2 = i_4$, and in this case it is exactly the variance of the random variable $\frac{(WX)_{ij}}{\sqrt{n_0}}$. □

We now consider four random entries, and we compute

$$\frac{1}{n_0^2}\kappa\Big((WX)_{i_1i_2},(WX)_{i_3i_4},(WX)_{i_5i_6},(WX)_{i_7i_8}\Big).$$

We observe that the cumulant vanishes identically if any index appears exactly once by the independence property, and thus each $W$- and $X$-index must appear exactly twice. This is only possible if we have two cycles on two indices each, or a single four-cycle. The cumulant of the former vanishes identically by independence ant thus the only non-vanishing 4-cumulant is

$$\kappa\left(\frac{(WX)_{i_1i_2}}{\sqrt{n_0}}, \frac{(WX)^*_{i_2i_3}}{\sqrt{n_0}}, \frac{(WX)_{i_3i_4}}{\sqrt{n_0}}, \frac{(WX)^*_{i_4i_1}}{\sqrt{n_0}}\right)$$

$$= \frac{1}{n_0^2}\mathbf{E}(WX)_{i_1i_2}(WX)^*_{i_2i_3}(WX)_{i_3i_4}(WX)^*_{i_4i_1}$$

$$= \frac{1}{n_0^2}\sum_{k_1,k_2,k_3,k_4=1}^{n_0}\mathbf{E}W_{i_1k_1}X_{k_1i_2}W_{i_3k_2}X_{k_2i_2}W_{i_3k_3}X_{k_3i_4}W_{i_1k_4}X_{k_4i_4}$$

$$= \frac{1}{n_0^2}\sum_{k_1=1}^{n_0}\mathbf{E}W_{i_1k_1}^2X_{k_1i_2}^2W_{i_3k_1}^2X_{k_1i_4}^2 = \frac{\left(\sigma_w^2\sigma_x^2\right)^2}{n_0}$$

Here for the first equality we used (14) where all but the trivial partition vanish identically since in some expectation a single index appears. This result can be generalised:

**Lemma A.3.** *For* $k \geq 2$ *and pairwise distinct indices we have*

$$\kappa\left(\frac{(WX)_{i_1i_2}}{\sqrt{n_0}}, \frac{(WX)^*_{i_2i_3}}{\sqrt{n_0}}, \frac{(WX)_{i_3i_4}}{\sqrt{n_0}}, \dots, \frac{(WX)^*_{i_{2k}i_1}}{\sqrt{n_0}}\right) = \frac{\left(\sigma_w^2\sigma_x^2\right)^k}{n_0^{k-1}} + \mathcal{O}(n_0^{-k}).$$

*Proof.* As illustrated for the case with four random variables, to have a nonzero cumulant, we can encode the $2k$ random variables as a cycle graph of length $2k$. Then, the only contribution comes from

$$\kappa\left(\frac{(WX)_{i_1 i_2}}{\sqrt{n_0}}, \ldots, \frac{(WX)^*_{i_{2k} i_1}}{\sqrt{n_0}}\right) = \frac{1}{n_0^k}\,\mathbf{E}(WX)_{i_1 i_2}\cdots(WX)^*_{i_{2k} i_1} = \frac{(\sigma_w^2 \sigma_x^2)^k}{n_0^{k-1}} + \mathcal{O}(n_0^{-k}),$$

which completes the proof. $\qquad\square$

Finally, we compute the cumulants of the entries of the random matrix $Y$. Since the activation function $f$ is applied component-wise, it follows from the previous results that the only contribution comes from $\kappa(Y_{i_1 i_2}, Y^*_{i_2 i_3}, Y_{i_3 i_4}, \ldots, Y^*_{i_{2k} i_1})$ for $k \geq 1$ and $i_1, \ldots, i_{2k}$ distinct, thus proving that $Y$ has cycle correlations.

*Proof of Proposition 3.2.* From the Berry-Esséen Theorem it follows that

$$\kappa(Y_{ij}) = \mathbf{E}Y_{ij} = \int_{\mathbb{R}} f(x)\frac{e^{-x^2/2\sigma_w^2\sigma_x^2}}{\sigma_w\sigma_x\sqrt{2\pi}}\,\mathrm{d}x + \mathcal{O}(n_0^{-1/2})$$

$$= \int_{\mathbb{R}} f(\sigma_w\sigma_x x)\frac{e^{-x^2/2}}{\sqrt{2\pi}}\,\mathrm{d}x + \mathcal{O}(n_0^{-1/2}) = \mathcal{O}(n_0^{-1/2}),$$

and

$$\kappa(Y_{ij}, Y^*_{ji}) = (1 + \mathcal{O}(n_0^{-1/2}))\int_{\mathbb{R}} f^2(\sigma_w\sigma_x x)\frac{e^{-x^2/2}}{\sqrt{2\pi}}\,\mathrm{d}x = \theta_1(f)(1 + \mathcal{O}(n_0^{-1/2})),$$

since the random variables $(WX)_{ij}/\sqrt{n_0}$ are approximately centred Gaussian with variance $\sigma_w^2\sigma_x^2$. Let $k > 1$. Then, since $f$ is a smooth function with compact support, we have that $f$ is in $C^l$ for some integer $l > 1 + \frac{2k^2}{k-1}$. Using the Fourier inversion theorem, it follows that

$$f(x_1) = \frac{1}{2\pi}\int_{\mathbb{R}} \hat{f}(t_1)\,e^{it_1 x_1}\mathrm{d}t_1$$

$$= \frac{1}{2\pi}\int_{|t_1|\leq n_0^{\frac{k-1}{2k}}} \hat{f}(t_1)\,e^{it_1 x_1}\mathrm{d}t_1 + \frac{1}{2\pi}\int_{|t_1|> n_0^{\frac{k-1}{2k}}} \hat{f}(t_1)\,e^{it_1 x_1}\mathrm{d}t_1$$

$$= \frac{1}{2\pi}\int_{|t_1|\leq n_0^{\frac{k-1}{2k}}} \hat{f}(t_1)\,e^{it_1 x_1}\mathrm{d}t_1 + \mathcal{O}\left((n_0^{\frac{k-1}{2k}})^{1-l}\right),$$

where we used $|\hat{f}(t_1)| \leq \frac{c}{(1+|t_1|)^l}$, for some positive constant $c$. For notational simplicity we work in the case $k = 2$, but the argument when $k > 2$ is the same. We compute

$$\kappa(Y_{i_1 i_2}, Y^*_{i_2 i_3}, Y_{i_3 i_4}, Y^*_{i_4 i_1})$$
$$= \frac{1}{(2\pi)^4}\int_{\forall i,\,|t_i|\leq n_0^{\frac{1}{4}}} \hat{f}(t_1)\hat{f}(t_2)\hat{f}(t_3)\hat{f}(t_4)\kappa(e^{it_1 Z_{i_1 i_2}}, e^{it_2 Z^*_{i_2 i_3}}, e^{it_3 Z_{i_3 i_4}}, e^{it_4 Z^*_{i_4 i_1}})\,\mathrm{d}\boldsymbol{t} + \mathcal{O}(n_0^{-2}),$$

$$= \frac{1}{(2\pi)^4}\sum_{l_1,\ldots,l_4\geq 1}\int_{\forall i,\,|t_i|\leq n_0^{\frac{1}{4}}} \prod_{i=1}^4\left(\hat{f}(t_i)\frac{(it_i)^{l_i}}{l_i!}\right)\kappa((Z_{i_1 i_2})^{l_1}, (Z^*_{i_2 i_3})^{l_2}, (Z_{i_3 i_4})^{l_3}, (Z^*_{i_4 i_1})^{l_4})\,\mathrm{d}\boldsymbol{t} + \mathcal{O}(n_0^{-2})$$

where we introduced $Z := WX/\sqrt{n_0}$ and in the second equality used that any cumulant involving the deterministic 1 vanishes identically. We now expand the cumulant involving powers of $Z$ via the well known formula [21, Theorem 11.30] in terms of partitions of the set $\{1, \ldots, l_1 + l_2 + l_3 + l_4\}$ whose joint with the partition $\{\{1, \ldots, l_1\}, \ldots, \{l_1+l_2+l_3+1, \ldots, +l_1+l_2+l_3+l_4\}\}$ is the trivial partition. By the independence property it is clear that the leading contribution comes from those partitions with one block connecting one copy of each of $Z_{i_1 i_2}, Z^*_{i_2 i_3}, Z_{i_3 i_4}, Z^*_{i_4 i_1}$ and the remaining

blocks being internal pairings. Since for odd $l_i$ there are $l_1!! \cdots l_4!!$ such partitions it follows that

$$\kappa(Y_{i_1 i_2}, Y^*_{i_2 i_3}, Y_{i_3 i_4}, Y^*_{i_4 i_1})$$

$$= \frac{1}{(2\pi)^4} \sum_{\substack{l_1,\ldots,l_4 \geq 1 \\ l_i \text{ odd}}} \int_{\forall i,\, |t_i| \leq n_0^{\frac{1}{4}}} \prod_{i=1}^{4} \left( \hat{f}(t_i) \frac{(it_i)^{l_i}}{(l_i - 1)!!} \right) \kappa(Z_{i_1 i_2}, Z^*_{i_2 i_3}, Z_{i_3 i_4}, Z^*_{i_4 i_1})$$

$$\times \operatorname{Var}(Z_{i_1 i_2})^{(l_1 - 1)/2} \cdots \operatorname{Var}(Z^*_{i_4 i_1})^{(l_4 - 1)/2} \, \mathrm{d}\boldsymbol{t} + \mathcal{O}(n_0^{-3/2})$$

$$= \frac{\sigma_w^4 \sigma_x^4}{n_0} \frac{1}{(2\pi)^4} \sum_{k_1,\ldots,k_4 \geq 0} \int_{\forall i,\, |t_i| \leq n_0^{\frac{1}{4}}} t_1 t_2 t_3 t_4 \prod_{i=1}^{4} \left( \hat{f}(t_i) \frac{(-\sigma_w^2 \sigma_x^2 t_i^2 / 2)^{k_i}}{k_i!} \right) \mathrm{d}\boldsymbol{t} + \mathcal{O}(n_0^{-3/2})$$

$$= \frac{1}{n_0} \left( \sigma_w \sigma_x \frac{1}{2\pi} \int \widehat{f'}(t) e^{-\sigma_w^2 \sigma_x^2 t^2 / 2} \, \mathrm{d}t \right)^4 + \mathcal{O}(n_0^{-3/2}),$$

where in the penultimate step we used Lemmata A.2–A.3 and in the ultimate step we used the Fourier property $\widehat{f'}(t) = it\hat{f}(t)$. Together with

$$\frac{\sigma_w \sigma_x}{2\pi} \int \widehat{f'}(t) e^{-\sigma_w^2 \sigma_x^2 t^2 / 2} \, \mathrm{d}t = \frac{1}{\sqrt{2\pi}} \int f'(x) e^{-x^2 / 2\sigma_w^2 \sigma_x^2} \, \mathrm{d}x$$

$$= \sigma_w \sigma_x \int f'(\sigma_w \sigma_x x) \frac{e^{-x^2/2}}{\sqrt{2\pi}} \, \mathrm{d}x = \theta_2(f)^{1/2}.$$

we conclude

$$\kappa(Y_{i_1 i_2}, Y^*_{i_2 i_3}, Y_{i_3 i_4}, Y^*_{i_4 i_1}) = \theta_2(f)^2 n_0^{-1} \left( 1 + \mathcal{O}(n_0^{-1/2}) \right),$$

just as claimed. $\qquad \square$

# B  Proof of Theorem 2.5

## B.1  Derivation of the self-consistent equation

We proceed as in Subsection A.1. We know from (15) that

$$\frac{1}{m} \sum_{i=1}^{m} \left( \frac{Y^* G Y}{m} \right)_{ii} = \frac{n_1}{m} \left\langle \frac{Y Y^* G}{m} \right\rangle = \frac{n_1}{m} (1 + zg). \tag{25}$$

We further claim the following.

**Lemma B.1.** *It holds that*

$$\frac{1}{m} \sum_{i=1}^{m} \sum_{j=1}^{n_1} \left( \frac{Y^* G Y}{m} \right)_{ij} = 1 + \mathcal{O}\left( (\theta_{1,b}(f)\, n_1)^{-1} \right). \tag{26}$$

Together with (25), Lemma B.1 implies

$$\frac{1}{m} \sum_{i \neq j} \left( \frac{Y^* G Y}{m} \right)_{ij} \approx 1 - \frac{n_1}{m} (1 + zg). \tag{27}$$

*Proof.* Using the Woodbury matrix identity[3], we have

$$\frac{1}{m} \left( \frac{Y^* G Y}{m} \right) = \frac{1}{m^2} Y^* \left( \frac{Y Y^*}{m} - z \right)^{-1} Y = \frac{1}{m} + \frac{z}{m} \left( \frac{Y^* Y}{m} - z \right)^{-1},$$

---

[3]For $A \in \mathbb{R}^{n \times n}$, $C \in \mathbb{R}^{r \times r}$, $U \in \mathbb{R}^{n \times r}$ and $V \in \mathbb{R}^{r \times n}$ the *Woodbury matrix identity* is given by

$$(A + UCV)^{-1} = A^{-1} - A^{-1} U \left( C^{-1} + V A^{-1} U \right)^{-1} V A^{-1}.$$

which implies

$$\sum_{i,j} \frac{1}{m} \left( \frac{Y^*GY}{m} \right)_{ij} = \sum_{i,j} \frac{1}{m} \delta_{ij} + \sum_{i,j} \frac{z}{m} \left( \frac{Y^*Y}{m} - z \right)_{ij}^{-1} = 1 + \sum_{i,j} \frac{z}{m} \left( \frac{Y^*Y}{m} - z \right)_{ij}^{-1}.$$

So, we need to show that $\sum_{i,j} \frac{z}{m} \left( \frac{Y^*Y}{m} - z \right)_{ij}^{-1}$ is approximately zero. Let $e := \frac{1}{\sqrt{m}} [1 \; \cdots \; 1]^T$ be a normalized vector in $\mathbb{R}^m$. We then write

$$\sum_{i,j} \frac{z}{m} \left( \frac{Y^*Y}{m} - z \right)_{ij}^{-1} = z \langle e, \left( \frac{Y^*Y}{m} - z \right)^{-1} e \rangle.$$

It turns out that $e$ is approximately an eigenvector of $\frac{1}{m} Y^*Y$. Indeed, it holds that

$$\mathbf{E} \left( \frac{Y^*Y}{m} e \right)_i = \frac{1}{m\sqrt{m}} \sum_{j=1}^{m} \sum_{k=1}^{n_1} \mathbf{E} \, Y_{ik}^* Y_{kj} \approx m^{-1/2} n_1 \, \theta_{1,b}(f) = (n_1 \, \theta_{1,b}(f)) e_i.$$

Moreover, the variance is approximately $\mathcal{O}(n_1/m)$, which means that the standard deviation is of order 1, while the expectation of order $n_1$. Thus, $e$ is approximately an eigenvector of $\frac{1}{m} Y^*Y$ with eigenvalue $n_1 \theta_{1,b}(f)$. Since $\theta_{1,b}(f)$ is nonzero by assumption, we have that $e$ is approximately an eigenvector of the matrix $\left( \frac{Y^*Y}{m} - z\mathbf{1}_m \right)^{-1}$ with eigenvalue $(n_1 \theta_{1,b}(f) - z)^{-1}$, from which the result follows:

$$\left| \langle e, \left( \frac{Y^*Y}{m} - z \right)^{-1} e \rangle \right| \approx \left| (n_1 \, \theta_{1,b}(f) - z)^{-1} \right| \ll 1. \qquad \square$$

Given Lemma B.1 and Proposition 3.3, we can now prove the global law for the random matrix $M$ with the cycle correlations.

*Proof of Theorem 2.5.* Applying Proposition 3.3 to (16) and using the same power counting argument as in (21) we obtain

$$
\begin{aligned}
1 + zg &\approx \frac{1}{n_1 m} \sum_{i_1, i_2}^{*} \kappa(Y_{i_1 i_2}, Y_{i_2 i_1}^*) \, \partial_{Y_{i_2 i_1}^*} (Y^*G)_{i_2 i_1} + \frac{1}{n_1 m} \sum_{i_1, i_2, i_3}^{*} \kappa(Y_{i_1 i_2}, Y_{i_3 i_1}^*) \, \partial_{Y_{i_3 i_1}^*} (Y^*G)_{i_2 i_1} \\
&\quad + \frac{1}{n_1 m} \sum_{k \geq 2} \sum_{i_1, \ldots, i_{2k}}^{*} \kappa(Y_{i_1 i_2}, \ldots, Y_{i_{2k} i_1}^*) \, \partial_{Y_{i_2 i_3}^*} \cdots \partial_{Y_{i_{2k} i_1}^*} (Y^*G)_{i_2 i_1} \\
&\approx \frac{\theta_1(f)}{n_1 m} \sum_{i_1, i_2}^{*} \partial_{Y_{i_2 i_1}^*} (Y^*G)_{i_2 i_1} + \frac{\theta_{1,b}(f)}{n_1 m} \sum_{i_1} \sum_{i_2, i_3}^{*} \partial_{Y_{i_3 i_1}^*} (Y^*G)_{i_2 i_1} \\
&\quad + \frac{1}{n_1 m} \sum_{k \geq 2} \frac{\theta_2^k(f)}{n_0^{k-1}} \sum_{i_1, \ldots, i_{2k}}^{*} \partial_{Y_{i_2 i_3}^*} \cdots \partial_{Y_{i_{2k} i_1}^*} (Y^*G)_{i_2 i_1},
\end{aligned}
$$

(28)

where we omitted reference to $\mathbf{E}$ to simplify notation. Given Lemma A.1, we only need to compute $\partial_{Y_{i_3 i_1}^*} (Y^*G)_{i_2 i_1}$:

$$\partial_{Y_{i_3 i_1}^*} (Y^*G)_{i_2 i_1} = \sum_{j=1}^{n_1} \partial_{Y_{i_3 i_1}^*} (Y_{i_2 j}^* G_{j i_1}) \approx -G_{i_1 i_1} \left( \frac{Y^*GY}{m} \right)_{i_2 i_3},$$

where we omitted the contribution of $\partial_{Y^*_{i_3 i_1}} Y^*_{i_2 j}$ since it is very small. Plugging the partial derivatives into (28), we get

$$1 + zg \approx \frac{\theta_1(f)}{n_1 m} \sum_{i_1,i_2}^* G_{i_1 i_1} \left(1 - \left(\frac{Y^* GY}{m}\right)_{i_2 i_2}\right) - \frac{\theta_{1,b}(f)}{n_1 m} \sum_{i_1} \sum_{i_2,i_3}^* G_{i_1 i_1} \left(\frac{Y^* GY}{m}\right)_{i_2 i_3}$$

$$- \frac{1}{n_1 m} \sum_{k \geq 2} \frac{\theta_2^k(f)}{n_0^{k-1}} \sum_{i_1,\ldots,i_{2k}}^* \partial_{Y_{i_3 i_4}} \cdots \partial_{Y_{i_{2k-1} i_{2k}}} \left(\frac{GY}{m}\right)_{i_3 i_{2k}} G_{i_1 i_1} \left(1 - \left(\frac{Y^* GY}{m}\right)_{i_2 i_2}\right)$$

$$\approx \theta_1(f)g\left(1 - \frac{n_1}{m}(1 + zg)\right) - \theta_{1,b}(f)g\left(1 - \frac{n_1}{m}(1 + zg)\right)$$

$$- g\left(1 - \frac{n_1}{m}(1 + zg)\right) \sum_{k \geq 2} \frac{\theta_2^k}{n_0^{k-1}} \sum_{i_3,\ldots,i_{2k}}^* \partial_{Y_{i_3 i_4}} \cdots \partial_{Y_{i_{2k-1} i_{2k}}} \left(\frac{GY}{m}\right)_{i_3 i_{2k}},$$

where in the second step we used (25) and (27). Finally, by shifting the index in the summation and doing some simple bookkeeping, we have

$$1 + zg \approx (\theta_1 - \theta_{1,b})g\left(1 - \frac{n_1}{m}(1 + zg)\right) - \theta_2 \frac{n_1}{n_0} g(1 + zg)\left(1 - \frac{n_1}{m}(1 + zg)\right)$$

$$+ \theta_2(\theta_1 - \theta_{1,b} - \theta_2)\frac{n_1}{n_0} g^2 \left(1 - \frac{n_1}{m}(1 + zg)\right)^2,$$

which corresponds to the self-consistent equation (6) as $n_0, n_1, m \to \infty$, where $\theta_1$ is replaced by $\theta_1 - \theta_{1,b}$. In the same way as in the bias-free case, the concentration inequality of Lemma 3.4 can also be applied here, thereby concluding that $g$ is approximately equal to its mean with high probability. The first claim of Theorem 2.5 then follows. The second claim follows easily from Lemma B.1. Since $n_1 \theta_{1,b}(f)$ is approximately an eigenvalue of the random matrix $\frac{1}{m} Y^* Y$, and since the nonzero eigenvalues of $Y^* Y$ are the same as the one of $YY^*$, we have that $\lambda_{\max} \approx n_1 \theta_{1,b}(f)$ is an eigenvalue of $M$ located away from the rest of the spectrum (called *outlier*). This concludes the proof of Theorem 2.5. □

## B.2 Proof of Proposition 3.3

In light of the central limit theorem, in the asymptotic limit the random variables $\frac{(WX)_{ij}}{\sqrt{n_0}} + B_i$ are approximately normally distributed with zero mean and variance $\sigma_w^2 \sigma_x^2 + \sigma_b^2$. In contrast to the bias-free case, here we have two different nonzero second cumulants of the entries of the random matrix $\frac{WX}{\sqrt{n_0}} + B$, and therefore also of the $Y_{ij}$'s.

*Proof of Proposition 3.3.* The first identity follows in a straightforward manner by assumption (8):

$$\kappa(Y_{ij}) = \mathbf{E}Y_{ij} = \int_{\mathbb{R}} f(x) \frac{e^{-x^2/2(\sigma_w^2 \sigma_x^2 + \sigma_b^2)}}{\sqrt{2\pi(\sigma_w^2 \sigma_x^2 + \sigma_b^2)}} \, dx + \mathcal{O}(n_0^{-1/2}) = \mathcal{O}(n_0^{-1/2}).$$

For the second cumulant, we first compute

$$\kappa\left(\frac{(WX)_{i_1 i_2}}{\sqrt{n_0}} + B_{i_1}, \frac{(WX)_{i_3 i_4}}{\sqrt{n_0}} + B_{i_3}\right) = \mathbf{E}\left(\frac{(WX)_{i_1 i_2}}{\sqrt{n_0}} + B_{i_1}\right)\left(\frac{(WX)_{i_3 i_4}}{\sqrt{n_0}} + B_{i_3}\right)$$

$$= \frac{1}{n_0} \mathbf{E}(WX)_{i_1 i_2}(WX)_{i_3 i_4} + \mathbf{E}B_{i_1} B_{i_3}$$

$$= \delta_{i_1 i_3} \delta_{i_2 i_4} \sigma_w^2 \sigma_x^2 + \delta_{i_1 i_3} \sigma_b^2.$$

For $i_1 = i_3$ and $i_2 = i_4$, the cumulant $\kappa(Y_{i_1 i_2}, Y^*_{i_2 i_1})$ follows easily:

$$\kappa(Y_{i_1 i_2}, Y^*_{i_2 i_1}) = (1 + \mathcal{O}(n_0^{-1/2})) \int_{\mathbb{R}} f^2(x) \frac{e^{-x^2/2(\sigma_w^2 \sigma_x^2 + \sigma_b^2)}}{\sqrt{2\pi(\sigma_w^2 \sigma_x^2 + \sigma_b^2)}} \, dx = \theta_1(f)(1 + \mathcal{O}(n_0^{-1/2})).$$

On the other hand, for $i_1 = i_3$ and $i_2 \neq i_4$, to compute the cumulant $\kappa(Y_{i_1 i_2}, Y^*_{i_4 i_1})$, we need the characteristic function of $\frac{(WX)_{i_1 i_2}}{\sqrt{n_0}} + B_{i_1}$ and $\frac{(WX)^*_{i_4 i_1}}{\sqrt{n_0}} + B_{i_1}$ which turns out to be asymptotically

equal to

$$\exp\left(-\frac{\sigma_w^2\sigma_x^2 + \sigma_b^2}{2}(t_1^2 + t_2^2) - \sigma_b^2 t_1 t_2\right).$$

Now, we can compute the cumulant of $Y_{i_1 i_2}$ and $Y_{i_4 i_1}^*$:

$$\kappa(Y_{i_1 i_2}, Y_{i_4 i_1}^*) \approx \frac{1}{(2\pi)^2}\int_{\mathbb{R}^2} f(x_1)f(x_2)e^{-i\boldsymbol{t}\cdot\boldsymbol{x}}\exp\left(-\frac{\sigma_w^2\sigma_x^2 + \sigma_b^2}{2}(t_1^2 + t_2^2) - \sigma_b^2 t_1 t_2\right)\mathrm{d}\boldsymbol{t}\,\mathrm{d}\boldsymbol{x}$$

$$= \frac{1}{(2\pi)^2}\int_{\mathbb{R}^2} \hat{f}(t_1)\hat{f}(t_2)\exp\left(-\frac{\sigma_w^2\sigma_x^2 + \sigma_b^2}{2}(t_1^2 + t_2^2) - \sigma_b^2 t_1 t_2\right)\mathrm{d}t_1\,\mathrm{d}t_2,$$

where in the second step we applied the Fourier inversion theorem. We denote the covariance matrix $\Sigma$ by

$$\Sigma := \begin{pmatrix} \sigma_w^2\sigma_x^2 + \sigma_b^2 & \sigma_b^2 \\ \sigma_b^2 & \sigma_w^2\sigma_x^2 + \sigma_b^2 \end{pmatrix} \tag{29}$$

with determinant $\det(\Sigma) = \sigma_w^2\sigma_x^2(\sigma_w^2\sigma_x^2 + 2\sigma_b^2)$ and inverse matrix

$$\Sigma^{-1} = \frac{1}{\det(\Sigma)}\begin{pmatrix} \sigma_w^2\sigma_x^2 + \sigma_b^2 & -\sigma_b^2 \\ -\sigma_b^2 & \sigma_w^2\sigma_x^2 + \sigma_b^2 \end{pmatrix}.$$

Again applying the Fourier inversion formula, we obtain

$$\kappa(Y_{i_1 i_2}, Y_{i_4 i_1}^*) \approx \frac{1}{(2\pi)^2}\int_{\mathbb{R}^2}\hat{f}(t_1)\hat{f}(t_2)e^{-\frac{1}{2}\langle\boldsymbol{t}, \Sigma\boldsymbol{t}\rangle}\mathrm{d}\boldsymbol{t}$$

$$= \frac{1}{(2\pi)^2}\int_{\mathbb{R}^2} f(x_1)f(x_2)\frac{2\pi}{\sqrt{\det(\Sigma)}}e^{-\frac{1}{2}\langle\boldsymbol{x}, \Sigma^{-1}\boldsymbol{x}\rangle}\mathrm{d}\boldsymbol{x}$$

$$= \frac{1}{2\pi\sqrt{\sigma_w^2\sigma_x^2(\sigma_w^2\sigma_x^2 + 2\sigma_b^2)}}\int_{\mathbb{R}^2} f(x_1)f(x_2)e^{-\frac{1}{2}\langle\boldsymbol{x}, \Sigma^{-1}\boldsymbol{x}\rangle}\mathrm{d}\boldsymbol{x} = \theta_{1,b}(f),$$

where

$$e^{-\frac{1}{2}\langle\boldsymbol{x}, \Sigma^{-1}\boldsymbol{x}\rangle} = \exp\left(-\frac{(\sigma_w^2\sigma_x^2 + \sigma_b^2)(x_1^2 + x_2^2) - 2\sigma_b^2 x_1 x_2}{2\sigma_w^2\sigma_x^2(\sigma_w^2\sigma_x^2 + 2\sigma_b^2)}\right).$$

To complete the proof, it remains to compute the joint cumulant of $Y_{i_1 i_2}, Y_{i_2 i_3}^*, Y_{i_3 i_4}, \ldots, Y_{i_{2k} i_1}^*$ for $k > 1$ and $i_1, \ldots, i_{2k}$ distinct. For notational simplicity, we prove the statement for $k = 2$. First, we use the cumulant asymptotics in order to asymptotically compute the characteristic function. The cumulants have match those of the bias-free case, except for

$$\kappa\left(\frac{(WX)_{i_1 i_2}}{\sqrt{n_0}} + B_{i_1}, \frac{(WX)_{i_1 i_2}}{\sqrt{n_0}} + B_{i_1}\right) = \sigma_w^2\sigma_x^2 + \sigma_b^2.$$

In addition to all these cumulants, we also have

$$\kappa\left(\frac{(WX)_{i_1 i_2}}{\sqrt{n_0}} + B_{i_1}, \frac{(WX)_{i_4 i_1}^*}{\sqrt{n_0}} + B_{i_1}\right) = \kappa\left(\frac{(WX)_{i_2 i_3}^*}{\sqrt{n_0}} + B_{i_3}, \frac{(WX)_{i_3 i_4}}{\sqrt{n_0}} + B_{i_3}\right) = \sigma_b^2.$$

Therefore, the log-characteristic function is given by

$$-\frac{\sigma_w^2\sigma_x^2 + \sigma_b^2}{2}\sum_{i=1}^4 t_i^2 - \sigma_b^2(t_1 t_4 + t_2 t_3) + \sum_{n\geq 1}\frac{(-1)^{n-1}}{n}\left(\frac{(\sigma_w^2\sigma_x^2)^2}{n_0}\prod_{i=1}^4 t_i + \mathcal{O}(n_0^{-2})\right)^n$$

$$= -\frac{\sigma_w^2\sigma_x^2 + \sigma_b^2}{2}\sum_{i=1}^4 t_i^2 - \sigma_b^2(t_1 t_4 + t_2 t_3) + \log\left(1 + \frac{(\sigma_w^2\sigma_x^2)^2}{n_0}\prod_{i=1}^4 t_i + \mathcal{O}(n_0^{-2})\right),$$

for $t_1, t_2, t_3, t_4 \in \mathbb{R}$ such that $|t_i| < n_0^{1/4}$. We obtain the characteristic function by taking the exponential of the above expression. By the same argument as in the proof of Proposition 3.2, we

have

$$\kappa(Y_{i_1 i_2}, Y_{i_2 i_3}^*, Y_{i_3 i_4}, Y_{i_4 i_1}^*)$$

$$= \frac{1}{n_0} \left( \frac{\sigma_w^2 \sigma_x^2}{(2\pi)^2} \int \widehat{f'}(t_1) \widehat{f'}(t_2) \exp\left( -\frac{\sigma_w^2 \sigma_x^2 + \sigma_b^2}{2}(t_1^2 + t_2^2) - \sigma_b^2 t_1 t_2 \right) dt_1 dt_2 \right)^2 + \mathcal{O}(n_0^{-3/2})$$

$$= \left( \frac{1}{2\pi \sqrt{\sigma_w^2 \sigma_x^2 (\sigma_w^2 \sigma_x^2 + 2\sigma_b^2)}} \int f(x_1) f(x_2) e^{-\frac{1}{2}\langle \boldsymbol{x}, \Sigma^{-1} \boldsymbol{x} \rangle} d\boldsymbol{x} \right)^2$$

$$+ \frac{1}{n_0} \left( \frac{\sigma_w^2 \sigma_x^2}{2\pi \sqrt{\sigma_w^2 \sigma_x^2 (\sigma_w^2 \sigma_x^2 + 2\sigma_b^2)}} \int f'(x_1) f'(x_2) e^{-\frac{1}{2}\langle \boldsymbol{x}, \Sigma^{-1} \boldsymbol{x} \rangle} d\boldsymbol{x} \right)^2 + \mathcal{O}(n_0^{-3/2}),$$

where $\Sigma$ is the matrix defined by (29). It then follows that

$$\kappa(Y_{i_1 i_2}, Y_{i_2 i_3}^*, Y_{i_3 i_4}, Y_{i_4 i_1}^*) \approx \mathbf{E} Y_{i_1 i_2} Y_{i_2 i_3}^* Y_{i_3 i_4} Y_{i_4 i_1}^* - \mathbf{E} Y_{i_1 i_2} Y_{i_4 i_1}^* \, \mathbf{E} Y_{i_2 i_3}^* Y_{i_3 i_4}$$

$$= \theta_2(f)^2 n_0^{-1} \left( 1 + \mathcal{O}(n_0^{-1/2}) \right),$$

as desired. The proof for $k > 2$ is similar. $\qquad\square$

## C  Proofs of auxiliary results

*Proof of Lemma 3.1.* By applying the Fourier inversion theorem, we have

$$\mathbf{E} X_1 f(\boldsymbol{X}) = \frac{1}{(2\pi)^n} \int_{\mathbb{R}^n} \int_{\mathbb{R}^n} x_1 f(\boldsymbol{x}) e^{-i\boldsymbol{t}\cdot\boldsymbol{x}} \varphi_{\boldsymbol{X}}(\boldsymbol{t}) d\boldsymbol{x} \, d\boldsymbol{t},$$

where $\varphi_{\boldsymbol{X}}(\boldsymbol{t})$ is the characteristic function of the $n$-dimensional random vector $\boldsymbol{X}$. It holds that $\int_{\mathbb{R}^n} (-ix_1) f(\boldsymbol{x}) e^{-i\boldsymbol{t}\cdot\boldsymbol{x}} d\boldsymbol{x} = \partial_{t_1} \hat{f}(\boldsymbol{t})$. Then, it follows that

$$\mathbf{E} X_1 f(\boldsymbol{X}) = \frac{i}{(2\pi)^n} \int_{\mathbb{R}^n} \left( \partial_{t_1} \hat{f}(\boldsymbol{t}) \right) \varphi_{\boldsymbol{X}}(\boldsymbol{t}) d\boldsymbol{t}$$

$$= -\frac{i}{(2\pi)^n} \int_{\mathbb{R}^n} \hat{f}(\boldsymbol{t}) \left( \partial_{t_1} \varphi_{\boldsymbol{X}}(\boldsymbol{t}) \right) d\boldsymbol{t}$$

$$= -\frac{i}{(2\pi)^n} \int_{\mathbb{R}^n} \hat{f}(\boldsymbol{t}) \left( \partial_{t_1} e^{\log \varphi_{\boldsymbol{X}}(\boldsymbol{t})} \right) d\boldsymbol{t}$$

$$= -\frac{i}{(2\pi)^n} \int_{\mathbb{R}^n} \hat{f}(\boldsymbol{t}) \left( \partial_{t_1} \log \varphi_{\boldsymbol{X}}(\boldsymbol{t}) \right) \varphi_{\boldsymbol{X}}(\boldsymbol{t}) d\boldsymbol{t}.$$

Cumulants can also be defined in an analytical way as the coefficients of the $\log$-characteristic function

$$\log \mathbf{E} e^{i\boldsymbol{t}\cdot\boldsymbol{X}} = \sum_{\boldsymbol{l}} \kappa_{\boldsymbol{l}} \frac{(i\boldsymbol{t})^{\boldsymbol{l}}}{\boldsymbol{l}!}, \tag{30}$$

where $\sum_{\boldsymbol{l}}$ is the sum over all multi-indices $\boldsymbol{l} = (l_1, \dots, l_n) \in \mathbb{N}^n$. We note that $\kappa_{\boldsymbol{l}}(X_1, \dots, X_n) = \kappa(\{X_1\}^{l_1}, \dots, \{X_n\}^{l_n})$ means that $X_i$ appears $l_i$ times. One can prove that this definition of cumulants is equivalent to the combinatorial one given by 14 (see [24] for a proof). Using definition (30) results in

$$\partial_{t_1} \log \varphi_{\boldsymbol{X}}(\boldsymbol{t}) = i \sum_{\boldsymbol{l}} \kappa_{\boldsymbol{l}+\boldsymbol{e_1}} \frac{(i\boldsymbol{t})^{\boldsymbol{l}}}{\boldsymbol{l}!},$$

where $\boldsymbol{l} + \boldsymbol{e_1} = (l_1 + 1, l_2, \dots, l_n)$. Since $(i\boldsymbol{t})^{\boldsymbol{l}} \hat{f}(\boldsymbol{t}) = \widehat{f^{(\boldsymbol{l})}}(\boldsymbol{t})$, we finally obtain

$$\mathbf{E} X_1 f(\boldsymbol{X}) = \sum_{\boldsymbol{l}} \frac{\kappa_{\boldsymbol{l}+\boldsymbol{e_1}}}{\boldsymbol{l}!} \frac{1}{(2\pi)^n} \int_{\mathbb{R}^n} \widehat{f^{(\boldsymbol{l})}}(\boldsymbol{t}) \varphi_{\boldsymbol{X}}(\boldsymbol{t}) d\boldsymbol{t} = \sum_{\boldsymbol{l}} \frac{\kappa_{\boldsymbol{l}+\boldsymbol{e_1}}}{\boldsymbol{l}!} \mathbf{E} f^{(\boldsymbol{l})}(\boldsymbol{X}),$$

where we again applied the Fourier inversion formula. $\qquad\square$

*Proof of Lemma A.1.* Let $\Delta^{i,j}$ denote a $m \times n_1$ matrix such that $\Delta_{kl}^{i,j} = \mathbf{1}_{\{(i,j)=(k,l)\}}$. Then, applying the resolvent identity, we get

$$\frac{\partial G}{\partial Y_{ij}^*} = \lim_{\epsilon \to 0} \frac{\left(\frac{Y(Y^*+\epsilon\Delta^{i,j})}{m} - z\right)^{-1} - \left(\frac{YY^*}{m} - z\right)^{-1}}{\epsilon} = -\frac{GY\Delta^{i,j}G}{m}.$$

It follows that $\partial_{Y_{ij}^*} G_{ab} = -\left(\frac{GY}{m}\right)_{ai} G_{jb}$ for $1 \le a, b \le n_1$, $1 \le i \le m$, and $1 \le j \le n_1$. Therefore, we have

$$\partial_{Y_{i_2i_1}^*} (Y^*G)_{i_2i_1} = \sum_{j=1}^{n_1} \partial_{Y_{i_2i_1}^*} \left(Y_{i_2j}^* G_{ji_1}\right) = G_{i_1i_1}\left(1 - \left(\frac{Y^*GY}{m}\right)_{i_2i_2}\right),$$

which proves (3.6a). We now compute

$$\sum_{j=1}^{n_1} \partial_{Y_{i_2i_3}^*} \partial_{Y_{i_{2k}i_1}^*} \left(Y_{i_2j}^* G_{ji_1}\right) \approx -\sum_{j=1}^{n_1} \partial_{Y_{i_2i_3}^*} \left(Y_{i_2j}^* \left(\frac{GY}{m}\right)_{ji_{2k}} G_{i_1i_1}\right)$$

$$\approx -\left(\frac{GY}{m}\right)_{i_3i_{2k}} G_{i_1i_1} + \left(\frac{Y^*GY}{m}\right)_{i_2i_2} \left(\frac{GY}{m}\right)_{i_3i_{2k}} G_{i_1i_1},$$

where the approximation in the first line comes from the fact that the contribution of $\partial_{Y_{i_{2k}i_1}^*} Y_{i_2j}^*$ is very small and can therefore be neglected. Since the off-diagonals of the resolvent of random matrices are small if $\Im z \gg n_1^{-1}$, the partial derivative $\partial_{Y_{i_2i_3}^*} G_{i_1i_1}$ can be omitted. This justifies the second approximation. So, we obtain

$$\partial_{Y_{i_2i_3}^*} \cdots \partial_{Y_{i_{2k}i_1}^*} (Y^*G)_{i_2i_1} \approx -\partial_{Y_{i_3i_4}} \cdots \partial_{Y_{i_{2k-1}i_{2k}}} \left(\frac{GY}{m}\right)_{i_3i_{2k}} G_{i_1i_1}\left(1 - \left(\frac{Y^*GY}{m}\right)_{i_2i_2}\right),$$

which completes the proof of Lemma A.1. $\qquad\square$

# D   Concentration inequality

*Proof of Lemma 3.4.* Without loss of generality, it suffices to prove the statement w.r.t. $\mathbf{E}_X$ since by cyclicity the statement for $\mathbf{E}_W$ is analogous. We write $X = (\boldsymbol{x}_1, \ldots, \boldsymbol{x}_m)$ with $\boldsymbol{x}_k = (x_{1k}, \ldots, x_{n_0k})'$, and similarly, $Y = (\boldsymbol{y}_1, \ldots, \boldsymbol{y}_m)$. We denote by $\mathcal{F}_k$, $1 \le k \le m$, the filtration generated by $\{\boldsymbol{x}_l, 1 \le l \le k\}$ and by $\mathbf{E}_k[\cdot] := \mathbf{E}_X[\cdot \mid \mathcal{F}_k]$ the conditional expectation w.r.t. $\mathcal{F}_k$. Now, we decompose $g(z) - \mathbf{E}_X g(z)$ as a sum of martingale differences

$$D_k := \mathbf{E}_k \operatorname{Tr}(M - z\mathbf{1}_{n_1})^{-1} - \mathbf{E}_{k-1} \operatorname{Tr}(M - z\mathbf{1}_{n_1})^{-1}, \quad \text{for } k = 1, \ldots, m.$$

By construction, we have $\mathbf{E}_m \operatorname{Tr}(M - z\mathbf{1}_{n_1})^{-1} = \operatorname{Tr}(M - z\mathbf{1}_{n_1})^{-1}$ and $\mathbf{E}_0 \operatorname{Tr}(M - z\mathbf{1}_{n_1})^{-1} = \mathbf{E}_X \operatorname{Tr}(M - z\mathbf{1}_{n_1})^{-1}$. It then follows that

$$g(z) - \mathbf{E}_X g(z) = \frac{1}{n_1} \sum_{k=1}^{m} \mathbf{E}_k \operatorname{Tr}(M - z\mathbf{1}_{n_1})^{-1} - \mathbf{E}_{k-1} \operatorname{Tr}(M - z\mathbf{1}_{n_1})^{-1} = \frac{1}{n_1} \sum_{k=1}^{m} D_k.$$

Next, we define $M_k := M - \boldsymbol{y}_k\boldsymbol{y}_k^*$. We note that

$$\mathbf{E}_k \operatorname{Tr}(M_k - z\mathbf{1}_{n_1})^{-1} = \mathbf{E}_{k-1} \operatorname{Tr}(M_k - z\mathbf{1}_{n_1})^{-1},$$

since $M_k$ is independent of $\boldsymbol{y}_k$ and therefore is also independent of $\boldsymbol{x}_k$. So, we have

$$D_k = (\mathbf{E}_k - \mathbf{E}_{k-1})[\operatorname{Tr}(M - z\mathbf{1}_{n_1})^{-1} - \operatorname{Tr}(M_k - z\mathbf{1}_{n_1})^{-1}].$$

Then, by the Shermann-Morrison formula, we have

$$\left|\operatorname{Tr}(M - z\mathbf{1}_{n_1})^{-1} - \operatorname{Tr}(M_k - z\mathbf{1}_{n_1})^{-1}\right| = \left|\frac{\boldsymbol{y}_k^*(M_k - z\mathbf{1}_{n_1})^{-2}\boldsymbol{y}_k}{1 + \boldsymbol{y}_k^*(M_k - z\mathbf{1}_{n_1})^{-1}\boldsymbol{y}_k}\right|$$

$$\le \frac{|\boldsymbol{y}_k^*(M_k - z\mathbf{1}_{n_1})^{-2}\boldsymbol{y}_k|}{\Im(\boldsymbol{y}_k^*(M_k - z\mathbf{1}_{n_1})^{-1}\boldsymbol{y}_k)}$$

$$\le \frac{1}{\Im z},$$

where the last inequality follows from the resolvent identity:

$$\begin{aligned}
|\boldsymbol{y}_k^*(M_k - z\mathbf{1}_{n_1})^{-2}\boldsymbol{y_k}| &\le \boldsymbol{y}_k^*(M_k - z\mathbf{1}_{n_1})^{-1}(M_k - \bar{z}\mathbf{1}_{n_1})^{-1}\boldsymbol{y_k} \\
&= \frac{\boldsymbol{y}_k^*\left((M_k - z\mathbf{1}_{n_1})^{-1} - (M_k - \bar{z}\mathbf{1}_{n_1})^{-1}\right)\boldsymbol{y_k}}{2i\,\Im z} \\
&= \frac{\Im(\boldsymbol{y}_k^*(M_k - z\mathbf{1}_{n_1})^{-1}\boldsymbol{y_k})}{\Im z}.
\end{aligned}$$

Thus, $|D_k| \le 2(\Im z)^{-1}$, and so $g(z) - \mathbf{E}_X g(z)$ is a sum of bounded martingale differences. We can now apply the Burkholder's inequality which states that for $\{D_k, 1 \le k \le m\}$ being a complex-valued martingale difference sequence, for $p > 1$,

$$\mathbf{E}\left|\sum_{k=1}^m D_k\right|^p \le C\,\mathbf{E}\left(\sum_{k=1}^n |D_k|^2\right)^{p/2},$$

where $C$ is a positive constant depending on $p$. We refer to [5, Lemma 2.12] for a proof of this inequality. By choosing $p = 4$, we get

$$\begin{aligned}
\mathbf{E}_X\,|g(z) - \mathbf{E}_X g(z)|^4 &= \frac{1}{n_1^4}\,\mathbf{E}_X\left|\sum_{k=1}^m D_k\right|^4 \\
&\le \frac{1}{n_1^4}C\,\mathbf{E}_X\left(\sum_{k=1}^m |D_k|^2\right)^2 \\
&\le \frac{16\,C\,m^2}{n_1^4\,(\Im z)^4} = \mathcal{O}(n_1^{-2}(\Im z)^{-4}),
\end{aligned}$$

just as claimed. $\qquad\square$

# E   Complex case

**Remark E.1.** *We can also consider matrices $X \in \mathbb{C}^{n_0 \times m}$ and $W \in \mathbb{C}^{n_1 \times n_0}$ of complex random entries with zero mean and variance $\mathbf{E}|X_{ij}|^2 = \sigma_x^2$ and $\mathbf{E}|W_{ij}|^2 = \sigma_w^2$. Let $M = \frac{1}{m}YY^*$ with $Y = f\left(\frac{WX}{\sqrt{n_0}}\right)$, and let $f\colon \mathbb{C} \to \mathbb{R}$ be a real-differentiable function satisfying $\int_{\mathbb{C}} f(\sigma_w\sigma_x z)\frac{e^{-|z|^2}}{\pi}\mathrm{d}^2 z = 0$. Set $\theta_1(f) = \int_{\mathbb{C}} |f(\sigma_w\sigma_x z)|^2\,\frac{e^{-|z|^2}}{\pi}\mathrm{d}^2 z$. Then, it can be proved that the normalized trace of the resolvent of $M$ satisfies equation* (7).