# OpenReview forum: "Analysis of one-hidden-layer neural networks via the resolvent method"
_NeurIPS.cc/2021/Conference — NeurIPS 2021 Poster_

### Official Review · Reviewer_Xqoe · 2021-07-14

**Rating:** 7
**Confidence:** 3

**Summary:**

The authors study the asymptotic eigenvalue distribution of the post-activation Gram matrix in a one-hidden-layer neural network, in a regime where all matrix dimensions are growing proportionally. For input data with i.i.d. centered entries and a weight matrix with i.i.d. centered weights, the authors provide a new proof (under weaker distributional assumptions) of a result first due to Pennington and Worah that the asymptotic eigenvalue distribution is characterized by a quartic equation in its Stieltjes transform. The authors then provide an extension of this result to a model with i.i.d. Gaussian biases in its hidden layer. A qualitative difference in the setting with biases is that---in contrast to the bias-free setting, where special choices of activation function can allow the eigenvalue distribution to remain the same as that of the input Gram matrix, and hence remain constant across multiple layers---the authors show that the eigenvalue distribution with biases must necessarily evolve over layers (cf. Remark 2.6 and the discussion on page 7).

The proof is based on deriving the fixed-point equation for the expected Stieltjes transform using the cumulant-expansion generalization of Gaussian integration-by-parts, and explicitly analyzing the asymptotic mixed cumulants of the entries of the post-activation matrix. A technical challenge is to translate the more easily-obtained understanding of mixed cumulants of the pre-activation entries to the post-activations, and the authors accomplish this using a Fourier-analytic technique.

**Main Review:**

I think this is a nice contribution to this body of literature: To my knowledge, it is the first work to use this method of proof to study this type of non-linear matrix model. The results of Theorems 2.1 and 2.2 are perhaps not so new in the context of existing work, but the authors illustrate how this method can be used to obtain a new result in the form of Theorem 2.5. There is a high-level component of the proof that I don't understand---see below---but if this is a misunderstanding on my part and can be easily clarified or addressed, my opinion is that this work is above the bar for NeurIPS.

----------------------------------------

Major comment: In Propositions 3.2, 3.3, and Lemma A.3, what exactly is the claim that is being made on which mixed cumulants are 0 and which are non-zero, and how is this used on lines 371 and 456 in the proofs of the main results?

My confusion lies in e.g. the following example: Consider the 4th-order cumulant 1/n_0^2 kappa( (WX)_{1,2}, (WX)_{1,2}, (WX)_{1,2}, (WX)_{1,2} ) where all four entries of WX coincide. Is this case treated in the discussion preceding Lemma A.3, and is Lemma A.3 claiming that this cumulant is 0? My understanding is that it's not 0, but since (WX)_{1,2}/sqrt(n_0) converges to a Gaussian, this is asymptotically 0. However now consider kappa( Y_{1,2}, Y_{1,2}, Y_{1,2}, Y_{1,2} ) for the post-activations: Is Proposition 3.2 claiming that this is 0? Since Y_{1,2} = f((WX)_{1,2}/sqrt(n_0)) converges to a non-Gaussian limit, I guess this is non-zero even in the asymptotic limit. But then perhaps I don't understand how this mixed cumulant (and analogous terms for larger k) is treated in the second equality of line 371. My confusion is related to the sentence "Since the activation function f is applied component-wise, it follows from the previous results... that Y has cycle correlations" on lines 420-423; I'm not sure what is meant by this sentence.

A higher-level comment is that I feel a bit uneasy about the extent of casework that has been done---not only for Lemma A.3 as discussed above, but also in the "Proof of (20)" starting on line 495, and the proofs of Lemmas A.4 and D.1 in appendix D.

------------------------

Other minor comments:

(1) I don't know if it's fair to characterize the moment method as "more commonly used" (in the abstract and conclusion). For example Refs. [19] and [9] also used the resolvent method, and I think an appendix of Ref. [9] derived also the quartic equation of Theorem 2.2 using a resolvent approach (although using a different method to analyze the resolvent).
(2) Line 51, is it common to call gamma here the learning rate?
(3) Can the authors state more precisely the moment conditions needed for nu_1 and nu_2 at the bottom of page 3, for their result to hold.
(4) In Eqs. (18) and (20) of Propositions 3.2 and 3.3 the pattern of indices and conjugations in "..." are not clear, and it would be helpful to write out a few more terms.
(5) Should the approximations of Proposition 3.2 and 3.3 be interpreted for each fixed k as n -> infinity? In the analysis of Eq. (16) one needs to consider the infinite series in k for each fixed n---can the authors elaborate on how the error terms are treated in this series?

**Time Spent Reviewing:**

5

---

> ### Author Response · Authors · 2021-08-10
> **Response to reviewer Xqoe**
>
> We are grateful to the reviewer for his/her careful reading of our paper and his/her insightful comments. Below is our response to the issues raised in the review.
>
> We are sorry for the confusion and are grateful to the reviewer for raising this point. In fact, we are not claiming that something like the cumulant $\kappa(Y_{12},Y_{12},Y_{12},Y_{12})$ for the post-activation random variables is (approximately) zero. Instead, as commonly observed in random matrix theory, something like the fourth cumulant of a single entry $Y_{12}$ has no influence on the asymptotic behavior of the resolvent (it does have an influence on the fluctuation but in the present paper we do not consider the fluctuation). On a technical level this fact follows from a simple power counting argument. For example in equation (23), the contribution of the fourth cumulant of $Y_{12}$ comes with three derivatives. However, each derivative of $G$ gains a further factor of $m^{-1}$ which is why the term corresponding to the fourth cumulant in (23) is negligible compared to the second cumulant. As a consequence, the second equality in (23) is only an approximate equality up to sub-leading corrections. We are sorry for this confusion which we will address in a revised version by estimating the terms corresponding to the neglected cumulants in (23) and similarly in (31).
>
> Regarding the casework done for example in Appendix D: Our writing aimed at demonstrating the moment calculation for some simple examples which show the main mechanisms at work. However, upon reading the helpful comments by the reviewer, we agree that it would be good to accompany these examples by a general power counting argument. Roughly we show that mixed moments of (WX)_{i_1 i_2}, (WX)*_{i_2 i_3}, (WX)_{i_3 i_4}, (WX)*_{i_4 i_1} satisfy an asymptotic Wick theorem modified by the cycle contributions since other pairings not only involving squares and four-cycles are sub-leading. From this modified asymptotic Wick theorem the claim on the mixed cumulants follows directly. We will add such a more principled argument in a revised version of the manuscript.
>
>
> The reviewer also mentions some other comments. Here is a point-by-point response to the reviewer’s minor comments:
>
> (1) We agree with the reviewer that in recent years resolvents have been studied also in the ML literature. In the random matrix literature, the three methods to analyze resolvents are (a) the moment method, (b) the resolvent method via the Schur complement formula, and (c) the resolvent method via the Stein’s method (= cumulant expansion). In the recent years, (c) has become the most used method and as far as we know our manuscript is the first application of (c) to non-linear random matrices, which is what we meant by our comment. However, we will clarify that the Schur-complement approach has been used prior to our work.
>
> (2) We used the same definition of the ridge-regularized least squares loss function as in [6, 21]. In both papers, the learning rate is denoted by $\gamma$.
>
> (3) In the paper we implicitly assume existence of moments of any order. More precisely, we assume that there exists a sequence of constants $c_p, p\in\mathbb{N}$ such that $\mathbf{E} (|w_{ij}|^p + |x_{ij}|^p) \le c_p$. By standard cut-off arguments it seems that it would be sufficient to assume finiteness of the first five moments, but we did not aim towards optimizing moment conditions. In a revised version we will clarify that we assume finiteness of arbitrary moments and indicate how this assumption can be relaxed.
>
> (4) We apologize for the lack of clarity and thank the reviewer for pointing this out. We will clarify this by adding a few more terms inside the cumulants of Eqs. (18) and (20).
>
> (5) We agree that the use of $\approx$ in this context is too vague. In a revised version of the manuscript, we will make the error estimates in Propositions 3.2 and 3.3 explicit and show how the errors are estimated for the application within (23).

---

> > ### Comment · Reviewer_Xqoe · 2021-08-27
> > **response**
> >
> > Thanks for this response---they address my concerns adequately, and (if the paper is accepted) I do encourage the authors to provide these fuller details of the argument as discussed.

---

### Official Review · Reviewer_5v5G · 2021-07-18

**Rating:** 6
**Confidence:** 4

**Summary:**

This paper studies the spectral distribution of random matrix inspired by a one-hidden layer neural networks. In the first model, the matrix is obtained by applying a nonlinear function entrywise to the product of large rectangular matrices with zero mean IID entries. In the second model, a rank-one bias term is added before applying the non-linearity.  The main results  establish convergence of the emprical spectral distribution to a non-random limiting measure in the regime where the dimensions of the matrices scale linearly. This limiting measure is characterized in terms of the the second moments of the distributions on the matrices, the relative scaling of the dimensions, and certain properties of the non-linearity (described by Gaussian integrals).

While the result for the first model has been derived previously (using moment based methods), the result for the second model is novel. Furthermore, in comparison to the prior work, this paper uses the resolvent method, which requires an evaluation of the cumulants instead of the moments -- a potentially benefit of this approach is that is can be applied more broadly. There is some discussion about the implications of the bias term in the context of multilayer networks. Specially, a non-zero bias preclude the certain invariance properties seen in the setting without bias.

**Limitations And Societal Impact:**

I think the discussion is appropriate.

**Main Review:**

The high level ideas of this paper are clearly explained. Also, the contribution of the paper is clear in the context of previous work. While the basic idea of the resolvent method itself it not new, this paper shows how it can be used to simplify and generalize the analysis of a problem of current interest. Hence, I think this paper will be of interest to researchers working in this area.

On the technical side, I have some concerns about the lack of higher order moment assumptions on the entries of the matrices. Unless I missed something, the author(s) require only the exists of finite second moments. But clearly some more assumptions are needed justify the CLT statement in lines 125-127. The nonexistence of moment conditions is also potentially problematic later (e.g., in the proof of approximate equality in Proposition 3.2) where higher order terms are said to be negligible. Overall, I would not be surprised if the all of the analysis goes through under some tail conditions (such as the ones given in citation [6]).

There is also room to improve the clarity of the technical results. I list some points below

	-What does the notation less-than-approximately-equal-to mean?

	- What is the definition of 1_m? (I assume identity matrix)

	- In line 146 - 147  there are some further statements about the limiting spectral measure (e.g., the nonzero part is continuous). These statements do not seem to follow from Theorem 1 which simply asserts the existence of a limit so I am not sure where they come from.

	- In line 148 , the function g(z) is the Stieltjes transform of the empirical measure, not the limit mu, and so the wording is a bit confusing.

	- Remark 2.4 part (ii) says that Theorem 2.1 follows from Theorem 2.2. However, doesn't one also need to establish uniqueness of the solution (in the proper limiting sense)?

	- It would be useful to provide full details of the batch normalization used in Figure 3.


**Time Spent Reviewing:**

3

---

> ### Author Response · Authors · 2021-08-10
> **Response to reviewer 5v5G**
>
> We are grateful to the reviewer for his/her careful reading of our paper and his/her insightful comments. Below is our response to the issues raised in the review.
>
> We thank the reviewer for pointing out the missing of higher order moment conditions on the entries of the matrices $W$ and $X$. Implicitly we assume existence of moments of any order. More precisely, we assume that there exists a sequence of constants $c_p, p\in \mathbb{N}$, such that $\mathbf{E} (|w_{ij}|^p + |x_{ij}|^p) \le c_p$. By standard cut-off arguments it seems that it would be sufficient to assume finiteness of the first five moments, but we did not aim towards optimizing moment conditions. In a revised version we will clarify that we assume finiteness of arbitrary moments and indicate how this assumption can be relaxed.
>
> The reviewer also discusses the unclarity of some notations and technical results. Here is a point-by-point response to the reviewer’s comments:
>
>
> * We thank the reviewer for pointing out the lack of explanation of the notation “less-than-approximately-equal-to”. The notation $A \lesssim B$ means that there exists some numerical constant $c$ such that $A \le cB$. We will clarify this in a revised version.
>
> * Throughout our paper, we denote by $\mathbf{1}_m$ the m x m identity matrix. We will clarify this notation in a revised version of the paper.
>
> * We thank the reviewer for pointing this out. The fact that the limiting measure is absolutely continuous away from 0 follows from the explicit equation (1). Similarly, the fact that the measure is compactly supported follows from the fact that for large $z$ the equation (1) has real solutions, and therefore by Stieltjes inversion, the measure is compactly supported. We will provide a clarification of these points in a revised version.
>
> * Concerning the wording at line 148, we thank the reviewer for pointing this out. We will make the sentence clearer.
>
> * We thank the reviewer for highlighting this point concerning Remark 2.3 (ii). The limiting equation (1) does not have a unique solution in the upper-half plane. However, the equation is still stable in a small neighborhood of the solution and therefore Theorem 2.2 implies that $g$ is approximately given by a properly chosen solution of (1), and so convergence follows. We will clarify this issue in a revised version of the manuscript.
>
> * We are happy to include more details of the batch normalization used in Figure 3 in a final version of the paper. Essentially, we normalized each feature by its standard deviation in each layer.

---

### Official Review · Reviewer_o1S8 · 2021-07-19

**Rating:** 6
**Confidence:** 3

**Summary:**

This paper studies the limiting singular value distribution of non-linear random matrix of the form Y=f(WX), where W and X both consist of iid entries and f is applied entrywise, in the proportional growth asymptotics. The limiting law is characterized in terms of its Stieltjes transform as the solution to a self-consistency equation. The same result was obtained before by Pennington and Worah [21] in the Gaussian case and later extended by Benigni and Péché [6] to the general case. The difference is that the method in the current paper is resolvent-based. However, it is unclear what the advantage of this approach is. For example, is the result stronger than that of [6] in the sense of the required assumptions? The authors should clarify the main contribution of this work and what is new.



**Main Review:**

The paper is in general well-written, but I have some difficulty understanding the technical details and I felt a few steps are either missing or skipped.

- The proof starts with the cumulant expansion in (16) and then applies Prop 3.2. But Prop 3.2 only deals with three special cases of cumulants. Why the contribution of other configurations are negligible? It seems it is assumed the only non-negligible contributions are those from even cycles. For example, it is claimed on p. 15 of the supplement "Finally, we compute the cumulants of the entries of the random matrix Y. Since the activation function f is applied component-wise, it follows from the previous results that the only contribution comes from κ(Yi1, Yi2i3^*, ..., Yi2ki1^*) for k ≥ 1 and i 1 ,...,i 2k distinct, thus proving that Y has cycle correlations. " This is not clear to me why this is the case and what the *previous results* being referred to are. For example, it is clear that odd cycles can be ignored due to the zero-mean assumption. But how about four cycles with double edges, something like \kappa(Y_12,Y_12,Y_23,Y_23,Y_34,Y_34,Y_41,Y_41)?

- Along the same line, in (23), I cannot understand the second line. Note that here the kappa in the first line has k arguments and kappa in the second line has 2k arguments. So this is not a matter of counting. I presume at this point we are removing all other contributions except for even cycles. So this needs to formally justified.

- Also, from the writing it seems in the second line of (23) the indices i_1,...,i_{2k} are required to be distinct. What about the first line and also those summations in Lemma 3.1 and in (16)? None of the these are specified.

- Another crucial step that is not justified is the last step of the proof of Prop 3.2, p16, which claims that the cumulant of the four tuple \approx expectation of their product. Is this true for all other cumulants? In fact, \approx is used throughout the paper without definition.

Other comments:
- line 106: The author should clarify \kappa(Y_{i_1,i_2},Y_{i_2,i_3}^*,...,Y_{i_2k,i_1}^*). Do you mean
\kappa(Y_{i_1,i_2},Y_{i_2,i_3}^*,Y_{i_3,i_4}^*,Y_{i_4,i_5}^*,...,Y_{i_2k,i_1}^*) or \kappa(Y_{i_1,i_2},Y_{i_2,i_3}^*,Y_{i_3,i_4},Y_{i_4,i_5}^*,...,Y_{i_2k,i_1}^*)? I presume the latter.

- Existence and uniqueness of the solution to the self-consistency equation (1) is not discussed. Perhaps this is in the previous work [21,6]?
- In equation (1) the dummy variable m conflicts with the sample size m which is also used everywhere. One should consider changing the notations.
- It would be good to comment early the meaning of the integral parameters. For example, theta_1(f) is the variance of the entry of Y.
- It is not very clear what "invariance" means as mentioned in the abstract and also in the paper. Perhaps you meant the limiting spectral distributions of Y=f(WX) and X coincide?
- Lemma A.1. What is \mathbb{H}? Is it only used for complex valued W/X?

**Time Spent Reviewing:**

4 hours

---

> ### Author Response · Authors · 2021-08-10
> **Response to reviewer o1S8**
>
> We are grateful to the reviewer for his/her careful reading of our paper and for his/her insightful comments. Below is our response to the issues raised in the review.
>
> In the abstract and in the last paragraph of the Introduction (c.f. lines 95-119), we described the main contributions of the paper, what is new compared to previous works (c.f. [6, 21]) and the method we used. The first results in the paper, i.e., Theorem 2.1 and 2.2, are not new in the literature but were proven in [6,21] under different assumptions and with the method of moments (c.f. Remark 2.4). Instead of computing moments, we consider $Y$ as a random matrix with correlated entries, and essentially, we evaluate cumulants. As stated in the lines 15-17 and as explained in the lines 95-119, a potentially advantage of this approach is that one does not deal with the combinatorics which arise when computing moments, and one therefore expects that it can be applied more broadly, especially when the combinatorics from the moment method become intractable. In the context of the manuscript under review, the main advantage of the resolvent method is that it allows to include an additive bias without much additional effort. In a revised version we will clarify that for the bias-free case our proof has no significant advantage beyond requiring less combinatorics.
>
> The reviewer discusses the unclarity of some technical details. Here is a point-by-point response to the reviewer’s comments:
> * Propositions 3.2 and 3.3 deal with those cumulants giving a leading order contribution to the final result. However, the reviewer is completely right that we do not adequately justify the fact that the contribution from other cumulants are indeed negligible. So, for example, the first equality in (23) should only be an equality up to a negligible error. The fact that the other cumulants are in fact negligible is due to a simple power counting mechanism: Each derivative acting on a resolvent G yields a further factor of $m^{-1}$, so, for example, the third cumulant of a single entry $Y_{ij}$ gives a smaller contribution than the leading order contribution from its second cumulant (or variance). A similar mechanism is responsible for the case of a four cycle with double edges (as brought up by the reviewer) to be negligible. In a revised version we will explain this power counting principle and justify the first equality in (23) more carefully.
>
> * In the paper we indicated when the indices $i_1, …, i_{2k}$ are distinct (e.g. for equations (18), (19), (23)).  If it has not been specified, it means that they do not have to be distinct. We will clarify this in a future version of the paper.
>
> * We thank the reviewer for pointing this out. Indeed, the last step in the proof of Proposition 3.2 on page 16 has not been justified. This step follows from writing the cumulant as a sum of products of moments. Those terms with the expectation of a single copy of $Y$ are negligible due to the centering of $Y$ (c.f. Eq. (17)), while those terms with the expectation of a product of two $Y$’s are negligible due to the mismatching index allocation. We will provide a clarification of this last step in a revised version.
>
> * We agree that the use of $\approx$ in this context is too vague. In a revised version of the manuscript, we will make the error estimates explicit.
>
> The reviewer also addresses some other comments. Here is a point-by-point response to these reviewer’s comments:
> * We apologize for the lack of clarity and thank the reviewer for pointing this out. By $\kappa(Y_{i_1 i_2}, Y^\ast_{i_2 i_3}, …, Y^\ast_{i_{2k} i_1})$ we mean $\kappa(Y_{i_1 i_2}, Y^\ast_{i_2 i_3}, Y_{i_3 i_4}, Y^\ast_{i_4 i_5}, …, Y^\ast_{i_{2k} i_1})$. We will clarify this in a revised version of the paper.
> *We are not aware that existence or uniqueness of the self-consistent equation (1) is addressed in the previous works [21, 6]. We observed empirically that for certain choices of parameters the equation may in fact have two solutions in the upper half place, c.f. Remark 2.3(iii).
> *We agree with the suggestion of the reviewer of changing the notation to avoid any confusion between the variable $m$ of Eq. (1) and the sample size $m$. We will use the notation $g_\infty$ instead of $m$ to indicate the limit of $g$.
> *We thank the reviewer for this suggestion. We will explain the meaning of the integral parameters on page 2 in a revised version of the paper.
> *We use the term “invariance” to indicate that the eigenvalue distribution is preserved by the nonlinear transformation $f$. In other words, the singular value distribution of $Y=f(WX)$ is the same as the singular value distribution of $X$.
> *We denote by $\mathbb{H}$ the upper half plane, i.e. $\mathbb{H}=\\{z \in \mathbb{C} : \Im(z) \geq 0\\}$. This definition can be found on page 4, line 151.

---

### Official Review · Reviewer_HGSJ · 2021-07-19

**Rating:** 8
**Confidence:** 3

**Summary:**

This paper studies the limiting eigenvalue-distribution of the gram matrix $M$ associated with a random feature model, and extends current theory (Pennington and Worah 2017; Benigni and Péché 2019, etc.) to the case of multiple layers with additive layer-wise additive bias. Understanding the distribution of eigenvalues of the gram matrix useful for the understanding of the training dynamics and generalization in neural networks.



**Limitations And Societal Impact:**

Yes, this has been taken care of.

**Main Review:**

Strengths
---

- A stark contrast between the bias-free case and the case with additive bias is shown: in the former case, the eigenvalue-distribution of $M$ converges almost-surely to a compactly supported deterministic probability distribution, whose Stieltjes transform satisfies a certain fourth order equation. In the later case, there is no such convergence (if there is more than 1 hidden layer), irrespective the chose activation function.

- The main technique used to prove the theorems is the so-called "resolvent method". This is in contrast with the very combinatorial and cumbersome "moment method" used in prior works.

- The paper is well-written and easy to follow. Related works are discussed in great detail, and the authors' contributions are well-situated.

Questions
----
L220 "However, in the case of an additive bias no such convergence can be observed."  Is this a conjecture, an empirical observation, or is this rigorously proved somewhere in the paper ?

Minor Issues
-----
- The super-script numbering of the references is non-standard (page 10).


**Time Spent Reviewing:**

2

---

> ### Author Response · Authors · 2021-08-10
> **Response to reviewer HGSJ**
>
> We are grateful to the reviewer for his/her careful reading of our paper and his/her insightful comments. Below is our response to the issues raised in the review.
>
> The reviewer asked to clarify our claim in L220 that in the case of an additive bias we do not observe convergence towards the solution of Eq. (1). In the context of L220, this is a purely empirical observation. However, we also have theoretical evidence supporting this empirical observation since in light of Proposition 3.3 and Remark 2.6, irrespective of the chosen activation function, already the random matrix $Y^{(1)}$ has leading order correlations and therefore convergence of the spectral density to the solution of Eq. (1) is not expected beyond the first layer. We will clarify the empirical nature of the claim in L220 and the relation to our rigorous results in a revised version of the manuscript.
>
> We thank the reviewer for bringing the non-standard super-script number of references to our attention. We will change the reference numbering to a more standard format.

---

> > ### Comment · Reviewer_HGSJ · 2021-09-01
> > **Update**
> >
> > Thanks for the clarifications. I will conserve my score of 8. Thus I strongly recommend accepting the paper.

---

### Decision · Program_Chairs · 2021-09-27

**Decision:**

Accept (Poster)

**Comment:**

This paper considers the limited singular value distribution of a random feature model, which consists of an entrywise nonlinearity and a product of two i.i.d. matrices. In particular, the generalization performance of a single hidden layer neural network where the input data and hidden layer weights are i.i.d. random can be determined through this analysis. The authors show that the resolvent method can be used to simplify and generalize the existing techniques for this problem. The reviews all agreed that the paper contains strong theoretical results, only expressed minor concerns and suggestions, and recommended acceptance. As Reviewer o1S8 noted, adding a discussion of the advantage of the resolvent approach and clarifying the novelty of the results w.r.t. existing work (e.g., dealing with additive bias) will improve the paper. Please take into account the updated reviews when preparing the final version to accommodate the requested changes. Thank you for your submission to NeurIPS.